# Long Code Arena: a Set of Benchmarks for Long-Context Code Models

**Egor Bogomolov**[1,2], **Aleksandra Eliseeva**[1], **Timur Galimzyanov**[1], **Evgeniy Glukhov**[1],
**Anton Shapkin**[1], **Maria Tigina**[1], **Yaroslav Golubev**[1], **Alexander Kovrigin**[1],
**Arie van Deursen**[2], **Maliheh Izadi**[2], **Timofey Bryksin**[1]
[1]JetBrains Research, [2]Delft University of Technology
lca@jetbrains.com

## Abstract

The fields of code and natural language processing are evolving rapidly, with models becoming better at processing long context windows — supported context sizes have increased by orders of magnitude over the last few years. However, there is a shortage of comprehensive benchmarks for code processing that go beyond a single file of context, while the most popular ones are limited to a single method. With this work, we aim to close this gap by introducing Long Code Arena, a suite of six benchmarks for code processing tasks that require project-wide context. These tasks cover different aspects of code processing: library-based code generation, CI builds repair, project-level code completion, commit message generation, bug localization, and module summarization. For each task, we provide a manually verified dataset for testing, an evaluation suite, and open-source baseline solutions based on popular LLMs to showcase the usage of the dataset and to simplify adoption by other researchers. We publish the benchmark page on HuggingFace Spaces with the leaderboard, links to HuggingFace Hub for all the datasets, and link to the GitHub repository with baselines: `https://huggingface.co/spaces/JetBrains-Research/long-code-arena`.

## 1 Introduction

The Machine Learning for Software Engineering (ML4SE) domain has gained popularity over the recent years, with increasingly more powerful models for text and code processing becoming available. According to a recent survey [26], the most common ML4SE tasks studied in the literature are code generation, code completion, code summarization, and program repair. Unfortunately, the majority of the existing benchmarks for assessing ML4SE models have two major limitations: a short length of the available context and a limited resemblance of the practical use cases [24, 34].

Two common approaches in modern natural language processing (NLP) are retrieval-augmented generation [19] and utilization of long contexts [54]. Retrieval-augmented approaches [6, 31] can base their predictions on information from large corpora of data using various search techniques, while the development of new architectures [47, 18, 21] and techniques [12, 5] allows models to process tens of thousands of even millions of tokens. Both long-context and retrieval-augmented models can in theory utilize information from an entire software project. However, most existing ML4SE benchmarks operate with short code snippets — methods or at most files. For example, two most popular code generation datasets—HumanEval [8] and MBPP [4]—require models to process fewer than 1,000 tokens and generate a short function, usually no more than 100 tokens long.

A new direction of agentic ML4SE benchmarks requires models to work with long contexts: SWE-bench [32] and its variations [63, 61], Commit-0 [66], MLE-Bench [7], and others. Yet, as such benchmarks focus on agentic solutions, they require models to do function calling and planning as

well, not only processing of long contexts. This makes them less suited for evaluation of processing long context and evaluation of smaller models. Another type of existing ML4SE benchmarks that operates with long code sequences is code completion at the repository level [38, 64]. Unfortunately, the existing works do not account for the iterative nature of software development: while solving the code completion task in a single file, the benchmarks allow models to use the rest of the project without restrictions. At the same time, other parts of the project can be written after the studied file and utilize its contents, giving the model hints that will not be present in the practical use-case.

In this work, we present *Long Code Arena*, a suite of novel benchmarks for ML4SE models that cover six tasks: library-based code generation, CI builds repair, project-level code completion, commit message generation, bug localization, and module summarization. We design all the tasks and datasets in such a way that they require models to use information from a project module or the entire project to successfully complete the task, yet don't require complex multi-step interactions. For all the tasks, samples used for evaluation are rigorously filtered and then manually verified to ensure the best possible data quality. The data for all the tasks comes from open-source repositories with permissive licenses. We also provide baseline solutions for all the tasks based on popular models, although this work does not aim at solving the tasks — baselines are provided solely to aid future research. Further work is required to identify the best approaches to individual tasks and better collection strategies.

We open-source the implementations of baselines, code for evaluation, and all the datasets via GitHub and HuggingFace, with the links available from our HuggingFace Space: `https://huggingface.co/spaces/JetBrains-Research/long-code-arena`.

## 2 Long Code Arena Benchmarks

Long Code Arena is a suite of six benchmarks that cover different aspects of code processing: generation, repair, completion, summarization, processing diffs. For each task, we gather an evaluation dataset of around a hundred to a thousand examples that requires models to operate with source code at the scale of a module or an entire repository. For most tasks, we focus on Python code due to its popularity and to manually verify the correctness of the samples. However, the collection methodology for all the tasks allows extending the benchmarks with more languages in the future.

All the datasets we collect in Long Code Arena are based on data from open-source GitHub repositories — source code, commit history, issues, as well as build data from GitHub Actions. First, we extract a common corpus of repositories for further processing. To do so, we get the list of repositories via GitHub Search [11] that pass the following filters used in other works to ensure the quality of the data [33]: at least 1,000 commits, at least ten contributors, issues, and stars, at least 10,000 lines of code, not a fork, last commit after 01.06.2023, and a permissive license (we use the most popular permissive licenses [57] — MIT, Apache-2.0, BSD-3-Clause, and BSD-2-Clause). After the filtering, we are left with 4,343 repositories that we then download via GitHub API along with issues and pull requests data. For the CI builds repair task, we also retrieve GitHub Actions logs for some repositories, which we describe in Appendix C. The only task that we base on the existing dataset is commit message generation, for which we find samples with large commits and long commit messages in the recent CommitChronicle dataset [16].

After the initial data collection stage, we prepare evaluation datasets for each of the six tasks separately. For this, we apply further task-specific filters to the collected data, and then manually examine the samples to ensure their correctness. In the following two subsections, we present the task description, data collection methodology, and the conducted experiments for library-based code generation and project-level code completion. We choose these two tasks out of six as they require different kinds of models: while code generation expects (possibly large) instruction-tuned models, code completion requires smaller base models. The rest of the tasks put requirements on the models similar to those of code generation. For them, we provide the task descriptions in Section 2.3 and further discuss data collection and experiments for each task in-depth in the Supplementary Materials (Appendices C, D, E, and F) due to the tight space restrictions.

### 2.1 Library-based Code Generation

**Task description.** The first task we want to describe is a novel library-based code generation task. Given a task description and access to the contents of a software library, the model should generate a

single file that solves the task utilizing methods from the given library. The problem is motivated by the need of programmers to write code that utilizes the present dependencies and in-project APIs rather than adding new dependencies and increasing project complexity.

In contrast to library-based code generation, existing code generation benchmarks require models to produce self-sufficient code snippets, such as solutions to algorithmic problems [8, 4, 25], domain-specific code [36], one-liners [62], etc. Among the existing works, the setup of the library-based code generation task is similar to repository-level code completion benchmarks that evaluate API completion [38, 64]. Contrary to them, our benchmark requires models to generate an entire program based on an instruction in natural language instead of a single API call or a single line.

**Collection methodology.** To prepare the benchmark, we first extract usage examples from the Python projects that we collected by finding directories in the project roots that contain "examples" in their name. Such usage examples are provided by the library authors in order to show the capabilities and use cases of their libraries. After collecting the examples, we filter them as described in Appendix A.1, and get 150 files (usage examples) from 62 libraries, with each file heavily relying on the APIs of the respective project.

To create instructions, we first run the selected 150 files through GPT-4 [1], prompting it to generate an instruction for generating the respective file. This leaves us with step-by-step instructions that the LLM should follow to generate a script utilizing the library at hand. Then, we manually fix each instruction in order to reduce hinting to specific library methods and ensure its correctness.

To build contexts for generation, benchmark users have access to contents of the libraries that include on average 254 Python files with 2.5M characters and 2,242 unique class and method names. The respective medians are 164 files, 1.4M characters, and 1,412 names. Also, the libraries contain from 136 to 7,846 API names with mean and median being 2,242 and 1,412, respectively.

**Metrics.** To assess the usage of the respective library, we propose a metric called *API Recall*. We calculate it as the ratio of library-specific API calls (called functions, instantiated classes, used constants) made in the ground truth solution, that also appear in the generated program. For example, if the ground truth solution made 5 such calls and the model correctly guessed 3 of them, it will receive API Recall $= 60\%$. We treat APIs as library-specific if their name appears only in a single library among all Python repositories that we collected.

**Baselines.** We develop and evaluate baselines based on a range of popular LLMs. As baselines, we use models from OpenAI: GPT-3.5-turbo, GPT-4 [1], GPT-4o, GPT-4o-mini [43], reasoning models o1, o1-mini [44], and o3 [46]; from Anthropic: Claude-3.5-Sonnet, Claude-3.5-Haiku, Claude-3-Opus [2], Claude-3.7-Sonnet [3]; from Mistral: Mistral-7B [29] and Mixtral-8x7B [30]; from DeepSeek: V3 [13] and R1 [14]; Qwen2.5-Coder-32B [27], and three versions of Llama-3.1 [15] with 8B, 70B, and 405B parameters.

For the context, we provide models with the list of available APIs from the target library, without specifying which of them are library-specific, *i.e.,* unique to this library and being used to compute the metric. We do not provide implementations or usages for them, just names, as the full list of APIs from a library can overflow a context window of 32,000 tokens. We sort each API list according to BM-25 [48], treating the respective instruction for generation as a query. To compute the BM-25 score we split the names by snake_case and camelCase, remove punctuation from them, and turn them into lower case. Then, we evaluate each model with different lengths of context, providing 0, 20, 200, 2000, or all API names from the library at hand, and suggesting in the prompt that they may be helpful. When selecting the API names, we pick the ones with the highest BM-25 scores. Note that when provided with no context, the model will solely rely on its current knowledge of the library.

Table 1 shows the results of evaluation for the baselines. Firstly, when provided with no information about the given library aside from its name, Claude-3.7-Sonnet and DeepSeek-V3 show the best results by far with 47% and 45% API Recall, respectively. These two models demonstrate their coding capabilities and knowledge of the less popular libraries, with which other models struggle. Moreover, they further increase their quality to 51% when given access to all the API names from the library, showing the best quality of all evaluated models.

Interestingly, Llama-3.1-450B and GPT-4 perform with a similar quality, overcoming the newer GPT-4o. The models show memorization capabilities, as these libraries should have appeared in the training data. However, both Llama-3.1-405B and GPT-4 struggle to correctly identify useful

Table 1: API Recall of baselines for the library-based code generation task. Missing values are due to the context being longer than the supported context window size of the model. The right-most column shows the difference in quality between model working with no library-specific context and maximum context that fits into the model.

| | #APIs in the context | | | | | |
| | None | 20 | 200 | 2000 | All | $\Delta$ |
|---|---|---|---|---|---|---|
| Claude-3.7-Sonnet [2] | **0.47** | **0.46** | **0.50** | **0.50** | **0.51** | +0.04 |
| DeepSeek-V3 [13] | 0.45 | 0.44 | **0.50** | **0.50** | **0.51** | +0.06 |
| Claude-3-Opus [2] | 0.43 | 0.45 | 0.46 | 0.50 | 0.49 | +0.06 |
| o3 [46] | 0.39 | 0.39 | 0.46 | 0.49 | 0.49 | +0.10 |
| Claude-3.5-Sonnet [2] | 0.44 | 0.43 | 0.47 | 0.48 | 0.48 | +0.04 |
| o1 [44] | 0.29 | 0.28 | 0.36 | 0.44 | 0.44 | **+0.15** |
| GPT-4o [43] | 0.33 | 0.33 | 0.40 | 0.41 | 0.41 | +0.08 |
| Claude-3.5-Haiku [2] | 0.27 | 0.30 | 0.37 | 0.40 | 0.40 | +0.13 |
| GPT-4 [1] | 0.37 | 0.36 | 0.40 | 0.40 | 0.38 | +0.01 |
| DeepSeek-R1 [14] | 0.23 | 0.26 | 0.31 | 0.35 | 0.38 | **+0.14** |
| Qwen2.5-Coder-32B [27] | 0.29 | 0.31 | 0.38 | 0.38 | - | +0.09 |
| Llama-3.1-405B [15] | 0.36 | 0.36 | 0.38 | 0.39 | 0.37 | +0.01 |
| o1-mini [44] | 0.21 | 0.26 | 0.32 | 0.33 | 0.32 | +0.11 |
| gpt-4o-mini [43] | 0.15 | 0.20 | 0.31 | 0.31 | 0.31 | **+0.16** |
| GPT-3.5-turbo | 0.17 | 0.19 | 0.23 | 0.25 | - | +0.08 |
| Llama-3.1-70B [15] | 0.23 | 0.25 | 0.26 | 0.24 | 0.24 | +0.01 |
| Mistral-7B [29] | 0.07 | 0.13 | 0.20 | 0.18 | - | +0.11 |
| Mixtral-8x7B [30] | 0.11 | 0.13 | 0.19 | 0.14 | - | +0.03 |
| Llama-3.1-8B [15] | 0.10 | 0.14 | 0.17 | 0.12 | 0.13 | +0.03 |

APIs when provided with long lists of them: the models improve the quality by 3% when given up to 2,000 library APIs. Furthermore, at the full context both models get confused and only show minimal quality boosts. The results suggest that despite being technically able to use contexts beyond dozens of thousands of tokens, Llama-3.1-405B and GPT-4 cannot efficiently utilize them for code generation.

On the other hand, the recently introduced reasoning models show their superior ability to navigate long contexts. The models o3, o1, o1-mini, and DeepSeek-R1 do not show outstanding results when used without any information about the library: o3 is the only model among them to compete with other top-tier models. Yet, all the reasoning models exhibit 10-16% API Recall improvements when given the full list of library APIs. This suggests that reasoning models can identify the required API names more often than other models, while not being proficient in using the given libraries after the training stage.

Among the smaller models, Qwen-2.5-Coder-32B shows 38% API Recall when given 2,000 API names in the context. The model does so while heavily relying on the context, as suggested by the 9% difference in the results compared to the empty context. At 32 billion parameters, Qwen-Coder performs significantly better than the Llama-3.1-70B, despite being more than two times smaller. The Llama-3.1 family of models does not show good utilization of long context across all three evaluated model sizes. One possible reason for that is the lack of training on specialized code-related data, which was performed for Qwen-Coder.

Based on the conducted experiments with the baselines, we conclude that our benchmark is not being saturated with the modern models, and it can be used to assess their abilities in utilization of long contexts, while simultaneously tracking models' coding capabilities.

## 2.2 Project-Level Code Completion

**Task description.** The second task that we describe is project-level code completion, targeting the completion of single lines. We formulate the task as follows: given relevant information from the project, which we call *context*, and a prefix of the *completion file*, one needs to generate the next line in this file. While there exist other repository-level completion datasets [64, 38], we use project

history from Git to mimic the real-world use case and avoid possible data leakages between files that arise when files in the context are written after the completed file and rely on the completed code. On top of that, we introduce a fine-grained classification of the completed lines by the used APIs.

**Collection methodology.** To create the dataset, we process the collected Python projects, traversing their Git histories to collect commits that were done after 01.01.2022. We extract newly added files from them, filtering out files with fewer than 200 lines or more than 2,000 lines. To collect the context for each file, we checkout the respective parent commit and save the contents of all the code and text files (*e.g.,* build files, documentation), constituting the repository as it was when the commit was made. Each datapoint contains the file for completion, a list of lines to complete with their categories (see the categorization below), and a repository snapshot that can be used to build the context.

We split our dataset into four parts based on the total size of `.py` files in the repository snapshot. As the reference for such a division, we chose the CodeLlama model [49], which has a context window of size 16K and about three characters per token. Based on this, we have four sets of samples with the following limits on the total number of characters in the context `.py` files: *small-context set* from 0 to $16K \times 3 = 48K$ characters; *medium-context set* from 48K to 192K characters; *large-context set* from 192K to 768K characters; *huge-context set* from 768K characters. We downsample datapoints to five datapoints per repository, and the repositories to 75 per set to ensure data diversity. The sizes of the four sets are 144, 224, 270, and 296 datapoints, respectively.

For each datapoint, we also provide a list of lines for completion—35 lines on average—since evaluating a code model on every line of a file is extremely resource-consuming. Moreover, not all lines are equally hard to complete; *e.g.,* function declaration lines can be challenging due to uncertainty, whereas loop definition can be straightforward. Taking this into account, we introduce a classification of the code lines into six categories depending on the used functions and classes.

1. *infile* — lines that call functions/classes defined in the same file;

2. *committed* — lines that call functions/classes defined in other files in the commit introducing the completion file;

3. *inproject* — lines that call functions/classes defined in the snapshot of the project before the commit;

4. *common* — lines that contain common functions such as `main` or `get`;

5. *non-informative* — lines that are too short, too long, contain prints, etc. (see Appendix B.2 for the full definition);

6. *random* — the rest of the lines.

Our main focus is on the first three categories, as they definitely require the utilization of context to form a correct completion. While each line can fall into multiple categories based on the content, we only assign the "most difficult" category to each line in the following order (from difficult to easy): *committed*, *inproject*, *infile*, *common*. We then sample on average ten completion lines per datapoint for the first four classes and five lines per datapoint for non-informative and random classes. Thus, for each file in the dataset, we have multiple lines that the model should complete. The total numbers of completion lines are 4,686, 8,676, 9,631, and 9,810 for each of four sets, respectively.

**Metrics.** The main metric for the project-level code completion task is the exact match of generated lines per category. This is a proportion of correct predictions calculated separately for each of the categories. The prediction is correct if it matches the ground truth after removing leading and trailing whitespaces from both. Additionally, we compute models' perplexity on the completion file as a proxy metric to estimate how well the provided context from the repository allows to model the completion file.

**Baselines.** We use the dataset to evaluate how well pre-trained code LLMs can utilize context from the given repository. Here we provide the full evaluation results for CodeLlama-7B in Table 3 (see the online leaderboard[1] for other models).

We provide several context composers as baselines:

---

[1]Online leaderboard: `https://huggingface.co/spaces/JetBrains-Research/long-code-arena`

Table 2: The perplexity values for CodeLlama-7B with different context composers. The lower perplexity value suggests better modeling quality.

| Additional context | All files | | | Only Python files | | | Difference with FL |
|---|---|---|---|---|---|---|---|
| | 256 | 1,753 | 12,000 | 256 | 1,753 | 12,000 | |
| File-level (FL) | 1.849 | 1.849 | 1.849 | 1.849 | 1.849 | 1.849 | 0.000 |
| Naive | 1.798 | 1.788 | 1.761 | 1.788 | 1.760 | 1.677 | 0.172 |
| Path distance (PD) | 1.783 | 1.727 | **1.607** | 1.782 | 1.726 | **1.601** | 0.248 |
| Half hemory (HM) | 1.799 | 1.789 | 1.743 | 1.789 | 1.765 | 1.670 | 0.179 |
| HM + PD | 1.782 | 1.730 | 1.636 | 1.783 | 1.729 | 1.636 | 0.213 |
| File length | 1.797 | 1.784 | 1.742 | 1.792 | 1.774 | 1.708 | 0.141 |
| Imports First | 1.791 | 1.769 | 1.732 | 1.785 | 1.751 | 1.666 | 0.183 |
| Only declaration + PD$^2$ | 1.785 | 1.741 | 1.710 | 1.785 | 1.739 | 1.708 | 0.141 |

- *Naive composer* — all the files from the repository snapshot are concatenated into one string with no specific order.

- *Path distance composer* — the order of the files is defined by the distance between files in a project file tree: if the file from the repository is closer to the completion file, then its content is closer in the context.

- *File length composer* — the order of the files is defined by the length of a file: shorter files are closer to the completion file.

- *Half memory composer* — each line from the repository files is removed with a probability of $0.5$, and the order of the files is the same as in the naive composer.

- *Imports first composer* — the order of the files is defined by an import relation of first degree: if any project files are imported in the completion file, then these files are closer to the completion file.

- *Only declarations composer* — some project files are left only with declaration lines, so we keep only names from the repository files.

To compare different context composers, we compute model's perplexity on the completion file as a proxy for completion quality (lower perplexity should lead to better completions). We report results for CodeLlama-7B and the *medium-context* dataset in Table 2. We vary the number of context tokens coming from other repository files from 256 to 12,000 in order to check that the introduction of the context is indeed helpful. For all the evaluated context composers, we see that additional context helps, and Python files are more important for completion than the others (*e.g.*, files in other programming languages or docs). Out of the ones we evaluated, the composer based on *Path Distance* performs the best with 0.25 drop in perplexity compared to the usage of a single file, so we use *Path Distance* for further experiments. We leave further exploration of different context composers for future work.

Table 3 shows the Exact Match for CodeLlama-7B with *Path Distance* and *File-level* composers. As in the previous experiment, introduction of new context boosts the results across all datasets. We observe the biggest quality improvements for the *inproject* completions, as they require information from other project files to find relevant APIs. Completion for other line categories improves as well, as the model is able to find similar snippets of code already written in the project.

In Appendix B.3, we report more experiments that further investigate the impact of the context size on the completion quality and compare a wide range of models: CodeLlama-7B [49], DeepSeek-Coder (1.3B, 6.7B, 33B) [23], Llama (3.1-8B, 3.2-1B, 3B) [15], and Qwen2.5-Coder (0.5B, 3B, 14B, 32B) [27].

## 2.3 Other Tasks

Due to the lack of space, the thorough descriptions of the collected datasets and evaluated models for the rest of the tasks can be found in the Appendix, while we provide the task formulations below.

---

[2] We leave only declarations in all files except for one.

Table 3: Results of the project-level code completion for CodeLlama-7B. The metric is Exact Match for the generated line.

| Set | Context | *infile* | *inproject* | *committed* | *common* | *non-informative* | *random* | *all* |
|---|---|---|---|---|---|---|---|---|
| Small | File-level | 0.35 | 0.16 | 0.33 | 0.32 | 0.28 | 0.42 | 0.35 |
| | Path Distance 16K | 0.37 | 0.27 | 0.34 | 0.33 | 0.29 | 0.43 | 0.37 |
| | Difference | +6% | +**68**% | +3% | +3% | +2% | +2% | +5% |
| Medium | File-level | 0.37 | 0.32 | 0.38 | 0.31 | 0.31 | 0.50 | 0.39 |
| | Path Distance 16K | 0.43 | 0.49 | 0.42 | 0.44 | 0.44 | 0.58 | 0.49 |
| | Difference | +16% | +**53**% | +10% | +42% | +42% | +16% | +26% |
| Large | File-level | 0.36 | 0.29 | 0.39 | 0.34 | 0.30 | 0.44 | 0.35 |
| | Path Distance 16K | 0.46 | 0.44 | 0.55 | 0.46 | 0.42 | 0.54 | 0.47 |
| | Difference | +27% | +**52**% | +41% | +35% | +40% | +23% | +35% |
| Huge | File-level | 0.40 | 0.34 | 0.44 | 0.34 | 0.30 | 0.50 | 0.39 |
| | Path Distance 16K | 0.44 | 0.43 | 0.54 | 0.41 | 0.40 | 0.54 | 0.45 |
| | Difference | +10% | +26% | +22% | +20% | +**36**% | +8% | +17% |

**CI Build Repair** (see Appendix C) asks models to generate a patch that fixes a real-life issue in a CI setup. The minimal set of data for the task consists of a repository snapshot at the commit that caused the failure of the workflow and the logs of the failed step. The task can also be performed in a simplified *oracle* setup. In this case, we put a list of relevant files and code blocks—extracted from the ground truth commit—into the prompt. An important feature of this task is run-based evaluation: we utilize GitHub Actions [20] to run the generated fixes and assess their correctness.

**Commit Message Generation** (see Appendix D) for large commits requires a model to generate a natural language description of changes performed in a single commit. The changes can be represented in different ways — in various diff formats, as separate versions of each file before and after the changes took place, and others. Moreover, models can utilize information from unchanged project files to better understand how changes impacted the project. CMG is a well-established task in academic research [52] and a prominent feature in developer tools [10, 9], however, researchers often limit the scope to short diffs [16], leaving the performance on larger commits unexplored. Moreover, the quality of commit messages from open-source repositories—the most common data source—is notoriously mixed [56]. We bridge these two gaps with our novel CMG benchmark, manually curated and tailored for larger commits.

**Bug Localization** (see Appendix E) can be formulated as follows: given an issue with a bug description and a repository snapshot in a state where the bug is reproducible, identify the files within the repository that need to be modified to address the reported bug. Although this is a subset of the larger bug-fixing problem, partially covered by SWE-Bench [32], bug localization requires its own separate evaluation. This independent assessment can provide a better understanding of the various approaches and their efficiency in identifying the precise location of bugs within the large code bases.

**Module Summarization** (see Appendix F) tasks a model to write textual documentation based on the module's or project's source code and intent (a one-sentence description of the expected documentation content). This task greatly increases the context size available to the models compared to the existing benchmarks that cover method- or class-level summarization [28, 39, 41]. The source of inspiration for the module summarization task is the fact that large projects often include high-level materials, such as quick start guides, tutorials, module documentation, and usage instructions. The task aims to alleviate the time-consuming and routine process of creating these materials.

# 3 Results Across Multiple Tasks

In addition to using Long Code Arena as a set of independent benchmarks, it can be used to assess capabilities of models across multiple tasks. This can be done by assessing models' results on all tasks but code completion. We exclude code completion here as it mainly targets base versions of models, while other tasks expect instruction-tuned models. We conduct such assessment for a set of nine models evaluated on the five tasks: the family of Llama-3.1 models [15], reasoning models OpenAI-o1 [44] and DeepSeek-R1 [14], and proprietary LLMs Claude 3.5-Sonnet, Claude-3.5-Haiku [2], GPT-4o [43], and Gemini-1.5-Pro [55].

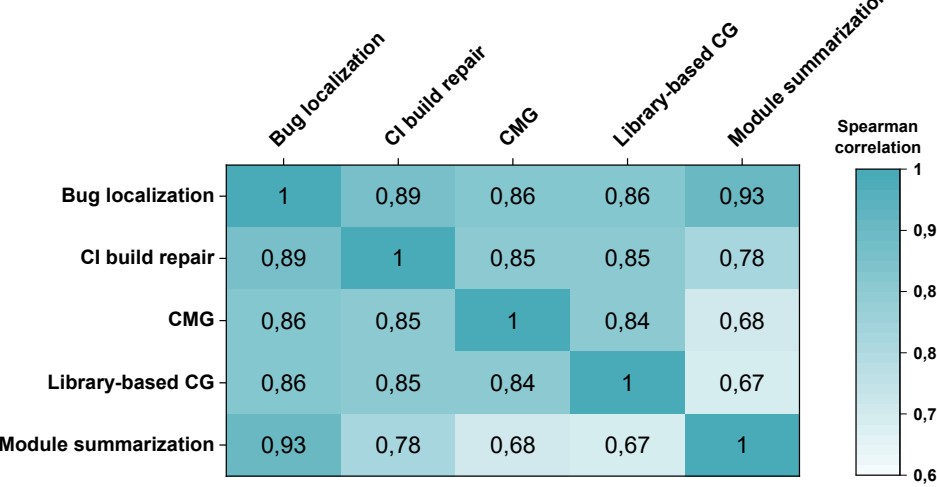

Figure 1: Correlation between models' results on the benchmarks.

Table 4 shows the results of models and their mean rank (from one to nine) across five tasks. To compute the mean ranks, we normalize the results across models for each task and treat the scores different by less than 10% as the same to reduce the effects due to randomness. o1 outperforms other models on all tasks but library-based code generation, where Claude-3.5 Sonnet shows slightly better results. The Llama-3.1 models lag significantly behind, despite the original report claiming the 405B version having coding and long context processing capabilities similar or better than Claude-3.5 Sonnet. We observe that the bug localization and module summarization are the tasks where reasoning models perform better, as these tasks require the most search capabilities. For module summarization, GPT-4o performs very well, which we attribute to its proficiency in writing long coherent texts. To further analyze task relations, we compute Spearman correlations between model scores on different tasks based on the common subset of models (see Figure 1). We observe high correlations between most tasks, which is expected given the wide gap in capabilities between some of the evaluated models. Yet, the correlations suggest that benchmarks are complementing each other.

## 4  Related Work

While there exist plenty of ML4SE datasets and even benchmark collections [40], most of them require models to operate with rather short contexts, around the size of a single method, which hinders the evaluation of novel long context models. Code generation datasets [8, 4, 37, 25, 22, 62] require models to process up to several paragraphs of the problem statement and then generate a short program (one line to one file). Existing datasets for code summarization [28, 40] target documentation in a

Table 4: Performance comparison across tasks for different models. BL: bug localization; CIR: CI build repair; CMG: commit message generation; LB-CG: library-based code generation; MS: module summarization.

| Model | Mean Rank | BL | CIR | CMG | LB-CG | MS |
|---|---|---|---|---|---|---|
| o1 | **1.0** | **0.58** | **0.24** | **36.4** | 0.45 | **70.9** |
| Claude-3.5 Sonnet | 1.6 | 0.52 | **0.24** | 34.8 | **0.48** | 66.1 |
| DeepSeek-R1 | 2.2 | 0.54 | 0.23 | 34.9 | 0.38 | 66.6 |
| GPT-4o | 2.8 | 0.53 | 0.10 | 34.8 | 0.41 | 67.0 |
| Gemini-1.5 Pro | 3.6 | 0.50 | 0.10 | 34.9 | 0.44 | 59.4 |
| Llama-3.1 (405B) | 5.2 | 0.47 | 0.04 | 34.8 | 0.37 | 59.6 |
| Claude-3.5 Haiku | 6.8 | 0.44 | 0.02 | 30.1 | 0.32 | 64.9 |
| Llama-3.1 (70B) | 7.0 | 0.35 | 0.05 | 33.5 | 0.24 | 58.5 |
| Llama-3.1 (8B) | 8.6 | 0.31 | 0.00 | 31.0 | 0.13 | 58.2 |

single method, meaning that both input and output size are below several hundred tokens. Previously developed commit message generation benchmarks [52, 16, 50] contain significantly shorter messages and diffs compared to Long Code Arena.

For code completion, recently, researchers introduced two benchmarks that operate at the repository scale: RepoEval [64] and RepoBench [38], also focusing on the completion of a single line. Compared to these benchmarks, we introduce a fine-grained classification of the completed lines and prevent possible data leakages by traversing Git history.

SWE-bench [32] and its extensions [63, 61] are recent benchmarks that require models to fix issues in real-world programming projects. Most solutions for these benchmarks use agentic approaches [60, 58, 65] which require models being compared to be capable of complex multi-turn interactions, planning, function calling. Long Code Arena covers a more diverse set of tasks, the most similar being CI builds repair, which focuses on builds in general rather than tests, and bug localization, which is a sub-task of the SWE-bench objective that we evaluate on a broader set of languages: Python, Java, and Kotlin. Yet, tasks in Long Code Arena are less restrictive for the models under evaluation and can distinguish between smaller models still being able to process long context windows.

The most notable benchmarks for long context models include Long Range Arena [53] and Scrolls [51]. Our work builds the first such benchmark focusing on ML4SE tasks, while Long Range Arena includes synthetic problems and Scrolls focuses on natural language processing.

# 5   Limitations and Future Work

In order to gather benchmarks for Long Code Arena, we had to make several design decisions that can impact the generalizability. First, we base the benchmarks on open-source data. This allows researchers to experiment with various context-collection techniques because they have access to source code data. On the other hand, modern LLMs use most available open-source data for training, and such reliance can lead to data contamination, which in turn can skew the evaluation results.

We argue that the tasks that we choose are less prone to models memorizing training data: there is no direct link between answers to benchmark tasks and raw repository data that modern models use for training. For example, while models could have seen documentation of specific libraries during training, currently it is unlikely that it was present side by side with the source code of the respective modules. The most memorization-prone task in our suite is code completion, but for it, we use historic data from Git repositories, which may become changed or overridden by the moment LLMs' training data is scraped.

In order to allow for manual examination of the collected data and to keep the benchmarks consistent, for most tasks we focus on datasets of Python code. Fortunately, the data preparation pipeline for all the tasks can be reused to produce datasets for other languages. The most complex step in this case will be manual verification and filtering of the data to ensure quality and correctness. In order to meet the quality requirement, we leave extension of datasets to other languages for future work.

In addition to extending datasets to other programming languages, future work includes collecting data for fine-tuning models for particular tasks and evaluating more models on the benchmarks. In order to assist other researchers with the latter, we open-source the code for the baseline solutions.

# 6   Conclusion

In this paper, we present the Long Code Arena. The goal of this work is to stimulate research in ML-based solutions for realistic software engineering tasks. In particular, we design a series of tasks that require taking a complex context into account, such as full projects, libraries and their usage, and coarse-grained components. Our work presents six benchmarks related to code generation, repair, completion, and summarization. For each task, we carefully design and manually curate evaluation data, metrics for assessing the results, and baseline solutions based on the pre-trained models. Our experiments show that the tasks are within reach, but far from solved. We hope and expect that our Long Code Arena will encourage researchers in ML4SE and NLP communities to advance the field of ML-enabled software engineering.

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

# Supplementary Materials

## A  Library-Based Code Generation

### A.1  Dataset Collection and Processing

The resulting dataset consists of 150 samples, each representing an instruction that a machine learning model should follow when generating a Python program, reference data for evaluation of the generation quality, and relevant data that can be used to improve generation. This relevant data is the source code of an entire Python library, based on a usage example from which we created the instruction for generation.

The structure of the individual datapoints is presented in Table 5. The labels are available in two forms: the reference program that was written by library authors as an example of library usage, and the list of library-specific API calls that the reference program makes. Both the program itself and the list of API calls can be used to assess the quality of a program generated by a machine learning model under evaluation. The dataset is self-contained, as it provides the snapshots of all associated repositories.

In order to collect the data, we use the following protocol:

1. We collect repositories from GitHub with at least 1,000 commits, at least ten contributors, issues, and stars, at least 10,000 lines of code, not a fork, last commit after 01.06.2023, and a permissive license (we use the most popular permissive licenses — MIT, Apache-2.0, BSD-3-Clause, and BSD-2-Clause). For the library-specific code generation task, we leave only repositories having Python as the main language.

2. For each repository, we detect the folder with usage examples: a folder with ".py" files that contains "examples" in its name. If a repository does not have such a folder, we filter it out. After this step, we are left with 883 repositories that have usage examples.

3. We then identify library-specific APIs for each of the 883 repositories. We extract all names of all methods, classes, and constants defined in these repositories, and treat as "library-specific" the ones that appear only in a single repository.

4. We then collect all Python files from the folders with examples and filter them: (i) remove examples shorter than 100 or longer than 40,000 characters (excluding comments), (ii)

Table 5: The structure of datapoints in the library-based code generation dataset.

| Field | Description |
| --- | --- |
| repo_full_name | Concatenated repository name and owner |
| repo_name | Library repository name |
| repo_owner | Library repository owner |
| instruction | Task for code generation |
| reference | Reference program written by the library authors |
| clean_reference | Reference program with comments removed |
| path_to_reference_file | Path to the reference in the repository (removed in repository snapshots to prevent data leakages) |
| path_to_examples_folder | Path to the directory with examples in the repository (removed in repository snapshots to prevent data leakages) |
| n_unique_apis | Number of calls to library-specific APIs in the reference program |
| unique_apis | List of calls to library-specific APIs in the reference program |
| project_defined_elements | All class and method names in the repository |
| api_calls | All API calls in the reference program |
| internal_apis | All API calls to the respective library in the reference program |

remove examples that have fewer than 400 characters of comments in order to then write high-quality instruction for generation, (iii) remove examples that use fewer than ten API calls specific to the given library. These filters result in 150 files (usage examples) from 62 libraries, with each file heavily relying on the APIs of the respective project.

5. After we have the usage examples for libraries, we create instructions for generating them. We first run the selected 150 files through GPT-4 [1], prompting it to generate an instruction for generating the respective file. You can see the prompt for generation in Figure 2. This leaves us with step-by-step instructions that the LLM should then follow to generate a script that utilizes the library at hand. Then, we manually fix each instruction in order to reduce hinting to specific library methods and ensure their correctness.

```
SYSTEM: We are developing a benchmark to assess quality of
code generation models. As a part of the benchmark, we include
the task of generating code based that uses the particular
library from a description in natural language. As a source of
data for this task we will use coding examples in Python
provided by library developers. Your task will be to generate
a text description of the provided Python code that will then
be used as an input for the generation task.

USER: Here is the code. You should write an instruction that
summarizes its contents and would allow another model to
generate this snippet of code, excluding the comments. Make
the instruction abstract, do not mention specific code
constructions that the generator should use. Be concise.
Generator will be able to access the contents of the following
library: [LIBRARY_NAME]. Use wording such as "Generate code
that ..." in your instruction.

[CODE]
```

Figure 2: Prompt for generating instructions from library usage examples.

# B  Project-Level Code Completion

## B.1  Datapoint Structure

Each instance that comprises the dataset consists of three key elements: a repository snapshot, a completion file, and target lines for the completion task. A repository snapshot is a list of all the filenames and contents of all text files from the repository (code, documentation, etc.). The state of the repository is before the commit where the completion file was added. A completion file is a Python file added in a particular commit. Target lines are a list of lines from the completion file that the model under evaluation should generate. Each line is also assigned one of classes that we describe in the following subsection.

The structure of datapoints:

- repo – repository name in the format {GitHub_user_name}__{repository_name}
- commit_hash – hash of the commit where the completion file was added
- completion_file – dictionary with the completion file content in the following format:
  - filename – path to the completion file
  - content – content of the completion file
- completion_lines – dictionary where keys are categories of lines and values are a list of integers (numbers of lines to complete). The categories are described in the following subsection.
- repo_snapshot – dictionary with a snapshot of the repository before the commit. Has the same structure as completion_file, but filenames and contents are organized as lists.

951 • `completion_lines_raw` – the same as `completion_lines`, but before sampling.

952 Targets for the completion task are provided in the `completion_lines` field. To get a target line for
953 completion, split the completion file by newline characters and select lines using the provided indices.
954 Line categories are also provided.

## B.2 Dataset Collection and Processing

956 Starting with the common corpus of repositories, we then follow the following process to acquire the
957 data:

1. **Traverse Git history**: We collect commits that add at least one new `.py` file. These files are
   candidates for the completion files.

2. **Filtering collected commits**: We filter the commits to retain only those with the potential
   completion files containing between 200 and 2,000 lines, and with creation dates after
   January 1st, 2022.

3. **Extract repository snapshots**: We create snapshots of the repositories based on the filtered
   commits, ensuring that we capture the state of the repository before the collected commit.
   The repository snapshots are intentionally not filtered to ensure that all possible information
   could be utilized. As a result, the dataset includes sources of noise, such as auto-generated
   files, CSV data, etc.

4. **Split by the size of relevant context**: We split all the data into four groups based on the
   number of characters in `.py` files from the repository snapshots. The groups are: (i) *small-
   context*: less than $48K$ characters; (ii) *medium-context*: from $48K$ to $192K$ characters;
   (iii) *large-context*: from $192K$ to $768K$ characters; (iv) *huge-context*: more than $768K$
   characters;

5. **Sample datapoints**: we randomly sample 5 datapoints for each repository, and we randomly
   sample 75 repositories for each group. If fewer than 5 datapoints or 75 repositories are
   available, we use all available datapoints or repositories. We keep all 80 repositories for the
   *medium-context* dataset.

6. **Classify lines**: We perform line classification that is introduced in the paper and assign a
   main category to each line of the completion file.

7. **Sample completion lines**: We sample lines from each category such that the average number
   of lines is no more than 5 for *non-informative* and *random* categories, and no more than 10
   for other categories.

982 Classification of the lines is done for each of the completion files. There are six categories of
983 completion lines according to various completion scenarios.

1. *infile* – a line contains at least one function or class that was declared in the completion file.

2. *inproject* – a line contains at least one function or class that was declared in the repository
   snapshot files.

3. *common* – a line contains at least one function or class that was classified to be common,
   *e.g.*, `main`, `get`, etc.

4. *committed* – a line contains at least one function or class that was declared in the files that
   were created in the same commit as the completion file (excluding the completion file).

5. *non-informative* – a line that satisfies at least on of the following criteria: (i) shorter than 5
   characters or longer than 150 characters, (ii) a line with `print`, (iii) a line with `import`, (iv)
   a declaration of a function or a class, (v) a comment or contains an inline comment.

6. *random* – all the lines that do not have any category.

995 Some lines may have more than one category after the classification. We additionally identify the
996 main category for each line based on the following approach.

997 • If a line has a *committed* category, then the main category is *committed*.

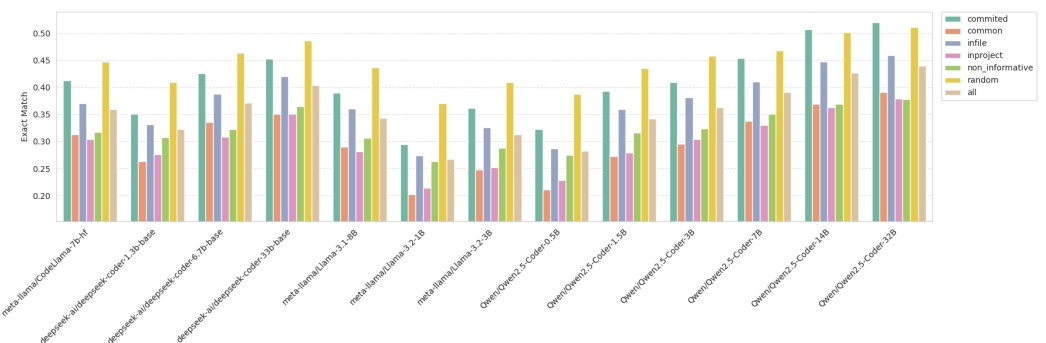

Figure 3: Large Context set, File-level.

- If a line does not satisfy the previous condition, but has an *inproject* category, then the main category is *inproject*.
- If a line does not satisfy previous conditions, but has an *infile* category, then the main category is *infile*.
- If a line does not satisfy previous conditions, but has a *common* category, then the main category is *common*.
- If a line has a *non-informative* category, then the main category is *non-informative*.
- If a line has a *random* category, then this is the only category for the line, and the main category is *random*.

The dataset has been collected in December of 2023. Considering the filtering process, the data within the dataset spans from January 2022 to December 2023.

We provide a distribution of lines for each set and each category in Table 6.

Table 6: Line counts for different sets in the project-level code completion dataset.

| Set | *infile* | *inproject* | *common* | *committed* | *non-informative* | *random* | *all* | **Avg. for one file** |
|-----|----------|-------------|----------|-------------|-------------------|----------|-------|----------------------|
| Small | 1,430 | 95 | 500 | 1,426 | 532 | 703 | 4,686 | 32.5 |
| Medium | 2,224 | 2,236 | 779 | 1,495 | 858 | 1,084 | 8,676 | 38.7 |
| Large | 2,691 | 2,595 | 693 | 1,322 | 1,019 | 1,311 | 9,631 | 35.7 |
| Huge | 2,608 | 2,901 | 692 | 1,019 | 1,164 | 1,426 | 9,810 | 33.1 |

## B.3 Extensive Evaluation

### B.3.1 Models Comparison

We compare a variety of models: CodeLlama-7B [49], DeepSeek-coder (1.3B, 6.7B, 33B) [23], Llama (3.1-8B, 3.2-1B, 3B) [15], and Qwen2.5-coder (0.5B, 3B, 14B, 32B) [27]. Comparison is made within the same setting: file-level completion, path distance composer with 16K context window, and the relative difference in Exact Match scores.

Figure 3 demonstrates that as the model size increases, performance metrics improve accordingly. Models effectively handle completion tasks across *random*, *committed*, and *infile* lines for the Large Context set. It is expected for *random* and *infile*, but it is unusual for *committed*. It could be an evidence that repositories from the large context set were in model's training data or that the committed API is too obvious.

Figure 4 shows that the Path Distance Composer enhances completion quality across all models, regardless of their family or size. The distribution of Exact Match scores per line category changes which supports our classification and the hypothesis behind it.

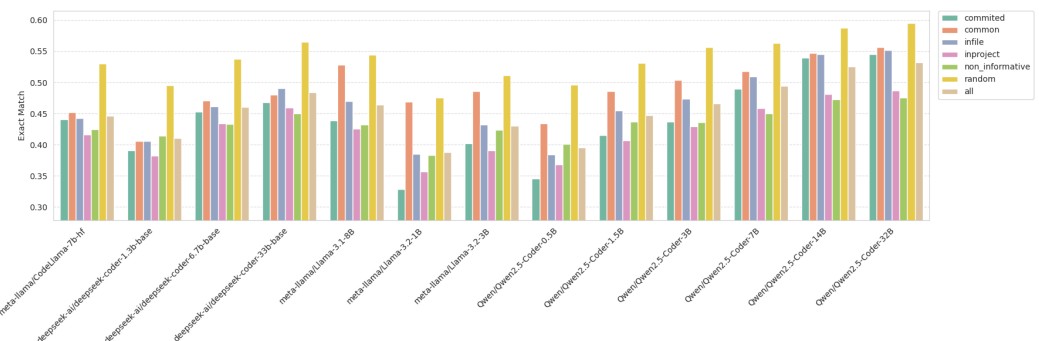

Figure 4: Large Context set, Path Distance composer, context window size is 16000.

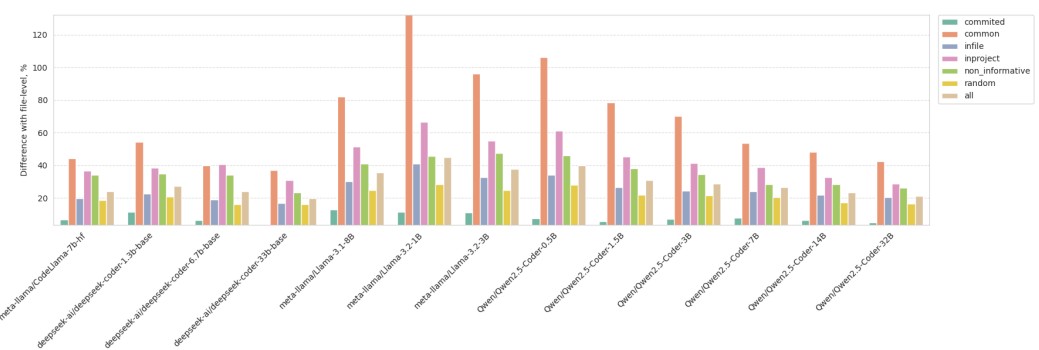

Figure 5: Large Context set, Difference between Path Distance 16K and File-level.

Figure 5 highlights the tendency that the bigger the model from a family the lower its completion quality gain from the context. That can be related to a fact that bigger models know more factual information, but smaller models successfully use in-context learning instead.

### B.3.2 Context Size Impact

We compare results of Qwen2.5-coder 7B on all the sets with different context window sizes: from 256 to 32000. Figure 6 illustrates that completion quality is better for a longer context across every line category. There are a few rapid shifts, e.g., *inproject* category for medium context set or *common* category for large context set. This behavior can be a result of a perfect file in the context.

An unexpected observation here is that *inproject* and *infile* categories improve with the same pace. So, the file-level information is not enough for the highest quality completion even for the *infile* lines.

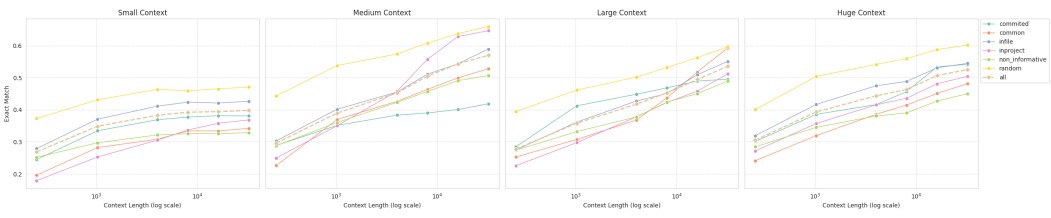

Figure 6: Qwen2.5-coder 7B, Path Distance context composer.

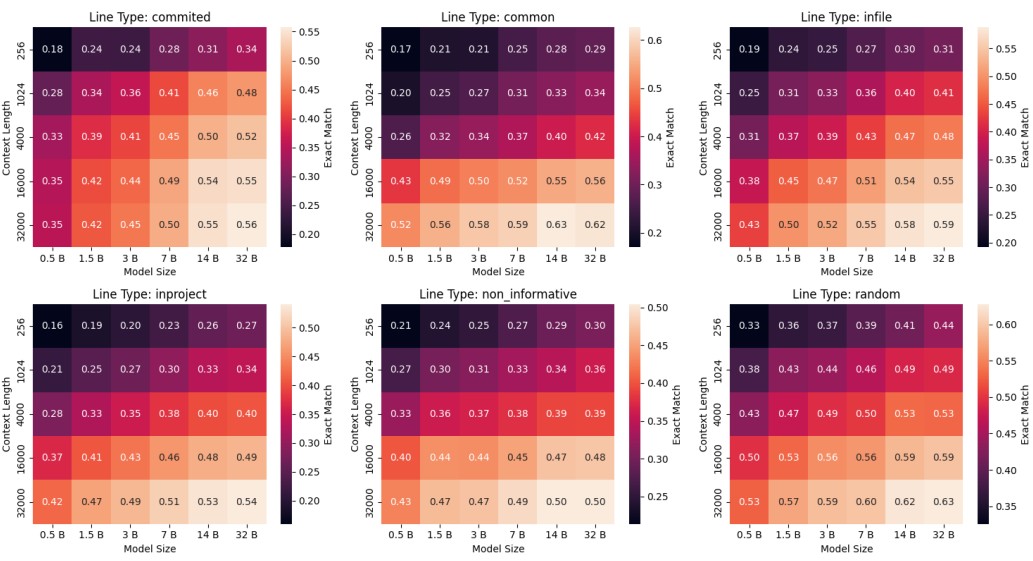

Figure 7: Qwen2.5-coder family of models with different context window sizes.

### B.3.3 Model Size vs Context Size

One of the possible applications of the presented dataset is to identify if the model size or context window matters the most. For example, Figure 7 shows that Qwen2.5-coder 32B with 32K context window performs almost the same as Qwen2.5-coder 14B with 32K context window, and Qwen2.5-coder 1.5B with 16K context window is equal to or better than any other Qwen2.5-coder model with 4K context window for most line types.

Overall, Figure 7 supports the general intuition that both context window size and model size positively impact performance. For the Qwen2.5-coder family, increasing both context length and model size leads to improved results across all task categories.

## C    CI Builds Repair

CI Build Repair asks models to generate a patch that fixes a real-life issue in a CI setup. The minimal set of data for the task consists of a repository snapshot at the commit that caused the failure of the workflow (*failed commit* hereafter) and the logs of the failed step. The task can also be performed in a simplified *oracle* setup by prompting a model with a list of files and their content or code blocks in them to change. In this case, the code blocks come from the ground-truth fixing diff provided in the dataset. An important feature of this task is run-based evaluation: we utilize GitHub Actions to run the generated fixes and assess their correctness.

### C.1    Dataset Collection and Processing

The final dataset consists of the datapoints with structure presented in Table 7. In order to collect and process the data, we use the following protocol:

1. We limited ourselves to the 100 largest Python repositories (main language: Python, the ratio of the main language $> 0.95$) with permissive licences. From each repository, we take no more than three branches, for each branch — no more than three different workflows, and for each workflow — no more than three datapoints. Thus, each repository could contribute up to 27 datapoints.

2. For all the collected Python repositories, we get the full list of the actions run in the repository, limited to last 90 days. Downloaded data contains action status (failed or successful) and links to the action runs.

Table 7: The structure of datapoints in the CI builds repair dataset.

| Field | Description |
| --- | --- |
| contributor | The username of the contributor that committed changes |
| difficulty | The difficulty of the problem according to an assessor on a 1–3 scale |
| diff | Contents of the diff between the failed and the successful commits |
| head_branch | Name of the original branch that the commit was pushed to |
| id | Unique ID of the datapoint |
| language | The main language of the repository |
| logs | List of dictionaries with logs of the failed job and name of the failed step in this job |
| repo_name | Name of the original repository |
| repo_owner | Owner of the original repository |
| sha_fail | SHA of the failed commit |
| sha_success | SHA of the successful commit |
| workflow | Contents of the workflow file |
| workflow_filename | The name of the workflow file (without full path) |
| workflow_name | The name of the workflow |
| workflow_path | The full path to the workflow file |
| changed_files | List of files changed in the diff |
| commit_link | URL to a commit corresponding to the failed job |

3. We gather a list of pairs of consecutive commits in which the first commit causes a failure of a workflow but the next one makes it build successfully.

4. For each pair of commits, we download:

   - logs of the failed step of the failed commit;
   - diff between the failed and successful commit (*correction diff*);
   - metadata of the failed commit.

   During the download, we clean the data according to the following filters (on the fly, to avoid excessive requests to GitHub API):

   - To reduce the benchmarking time, we eliminate runs that take more than 10 minutes (measured on successful action run).
   - To minimize the number of actions that contain pure formatting issues, we filter out datapoints, in which the names of the workflow, target, or failed step contain any of the following substrings: {*mypy*, *lint*, *flake8*, *black*}. We allow these substrings in the target name if there is more than one target in the action run.
   - We remove runs for which the workflow file contains substrings {*token*, *secret*} to ensure that we can run them without any prerequisites.
   - We keep only datapoints for which the correction diff (i) contains at least one .py file, and (ii) only contains files that match either of the following items: {code file, *.md*, *.rst*, *LICENSE\**, *readme\**, *doc/\**}. We do so to ensure that there are no changes in artifacts such as resources or data files, which the model cannot fix given the present context.

5. To isolate the problem to a single issue per datapoint, when running the benchmark, we delete all .yaml files in the .github/workflows/ directory, ensuring that only this workflow would be run. We also remove workflows that contain links to other workflow files to make sure that the target workflow is independent.

6. The human assessor assessed the datapoints to verify that logs contain all the necessary information to fix the issue and graded the datapoints on a 1–3 scale according to their difficulty. Table 8 describes the difficulty levels and the sizes of the available buckets.

7. In the last step, we run all datapoints through our benchmark at both the failed and the successful commit. We then keep only the datapoints that remained failing / passing at the

Table 8: Data split by the difficulty.

| Difficulty | # of datapoints | Description |
|:---:|:---:|:---|
| 1 | 36 | Issues with formatting |
| 2 | 7 | Local issues or issues with typing |
| 3 | 25 | Issues that require information about other files in the repository |
| **Total** | **68** | |

Table 9: Number of datapoints on each mining step.

| Data mining step | # of datapoints |
|:---:|:---:|
| Initial set of sampled workflows | 336 |
| Datapoints that passed assessor verification | 210 |
| Datapoints that passed GitHub Actions | 144 |
| Datapoints that passed GitHub Actions after 14 months | 68 |

respective commits. Moreover, we repeat the procedure after 14 months from the initial procedure to ensure the durability of the dataset. This last step is crucial as it filtered out 50% of the datapoints: quite many passing workflows started failing due to changes in library versions that were not specified by repository owners, connection issues, missing remote files or certificates. Table 9 reports the number of filtered datapoints at each step.

Context-related statistics are presented in Table 10

Table 10: Context-related statistics.

| Context metric | Mean | Median |
|:---:|:---:|:---:|
| Symbols in logs | 145K | 6.5K |
| Files in repository | 610 | 240 |
| Lines in repository | 170K | 56K |
| Symbols in repository | 7.5M | 2.4M |

## C.2 Evaluation

We implement the benchmark for using the CI builds repair dataset in our repository. The benchmark requires a user-implemented function (*fix_repo_function*) that repairs locally stored repository, given the logs of a failing build. The procedure is the following:

1. The benchmark clones each repository snapshot with depth equal to 1 to a local machine.

2. Then, the benchmark runs the model under evaluation, which takes a datapoint as input (mainly — log and workflow files) and needs to repair the repository on the local machine by editing or replacing files.

3. The benchmark edits the workflow files to run only one workflow.

4. Then, it pushes the current state of the repository to a new branch in the separate GitHub organization.

5. When results of builds in GitHub Actions become available, the benchmark collects, analyzes, and returns them.

To use the benchmark, one needs to send a request to join the GitHub organization[3] since the procedure requires pushing changes to repositories in that organization. Moreover, keeping repositories as forks in a separate organization ensures that they will remain available. The function *fix_repo_function* takes the following (all optional) arguments:

---

[3]GitHub Organization for the benchmark: `https://github.com/LCA-CI-builds-repair`

1115    1. **datapoint**: datapoint from the dataset

1116    2. **repo_path**: path to the repository on the user's machine

1117    3. **repo**: git.Repo object from the GitPython library

1118    4. **out_folder**: directory for outputting the benchmark results

1119 Intermediate results contain datapoint ID and meta information, as well as the SHA of the commit
1120 pushed to the target repository. After collecting the results, the benchmark adds the status of the
1121 GitHub Actions build to this information.

1122 We use the collected dataset to assess multiple LLMs in the CI builds repair task.

1123 To make the task easier to tackle, we provide models with an oracle — when asking to fix the build,
1124 we also provide the list of files and specific code blocks in them that should be fixed. The information
1125 on which files need fixing comes from the ground truth commit that fixed the build. In the future, if
1126 the task becomes too easy for the models, oracle can be simply removed to make the task even more
1127 realistic and challenging.

1128 To avoid compatibility issues with external packages, we implemented *time machine*, which ensures
1129 that installed package versions match those available at the time of the commit.

1130 To prompt the models to solve the task, we use the following strategy. To prepare an instruction,
1131 we locate the first occurrence of case-insensitive substring "error", "failure", "failed" or "traceback"
1132 in the logs and take a 200-line context around this occurrence (100 lines before and after). If the
1133 substring is not found, we use 200 last log lines. The instruction then reads as follows:

```
Title: Tests Failed After New Commit

## Overview
A recent commit caused one or more tests to fail in the repository.
We need to investigate the relevant logs, determine the problem, and propose a fix.

## Relevant Logs
Below is a focused snippet of the CI logs surrounding the failure:
```

1134 `{relevant_logs}`

1135 We then prompt the LLM to modify the code blocks provided by an oracle to align with the given
1136 instructions, and pass all the files in a single request in the following way:

```
[start of file.py]
...
```
1137 `[end of file.py]`

1138 LLM replies with a unified diff[4]. During evaluation of the benchmark results, these diffs are applied
1139 and the patched version is sent to GitHub Actions to be tested. The statistics of the context length
1140 (OpenAI models' tokens [45]) is following: min = 859, max = 61,982, mean = 13,994, std = 14,379,
1141 median = 9,726.

1142 Table 11 shows the evaluation results for three independent runs of several models: proprietary
1143 OpenAI GPT-4o [43], Anthropic Claude 3.5 Sonnet, 3 Opus, 3 Haiku [2], and Google Gemini 1.5
1144 Pro [55] (max context length = 32,768 due to technical reasons), as well as open-source DeepSeek-
1145 R1 [14] (max context length = 16384) and Llama instruct models [15]: INT8 Llama 3.1 (8B, 70B,
1146 405B). If not stated otherwise, all models have context length $\geq$ 64,000 tokens.

---

[4]Aider: `https://aider.chat/docs/unified-diffs.html`

Table 12: The structure of datapoints in the commit message generation dataset.

| Field | Description |
|-------|-------------|
| **repo** | The full name of the GitHub repository the commit comes from |
| **hash** | The SHA hash of the commit, serves as an identifier inside individual repository |
| **date** | The timestamp of the commit (from the commit author) |
| **license** | The type of the license in the repository of the commit |
| **message** | The ground truth commit message |
| **mods** | The changes performed in a commit, represented as a list of per-file modifications, where the structure of a per-file modification is described in Table 13 |

Table 11: Pass@1 scores of the CI builds repair benchmark for various LLMs. Average of three runs.

| Model | Pass@1, % |
|-------|-----------|
| DeepSeek-R1 | $23 \pm 1$ |
| Claude-3.5-Sonnet | $24 \pm 1$ |
| GPT-o1 | $19 \pm 1$ |
| Claude-3-Opus | $14 \pm 3$ |
| Claude-3-Haiku | $2 \pm 2$ |
| Gemini-pro-1.5 | $10 \pm 3$ |
| GPT-4o | $10 \pm 1$ |
| Llama-3.1-405B | $4 \pm 1$ |
| Llama-3.1-70B | $5 \pm 3$ |
| Llama-3.1-8B | $0$ |

# D    Commit Message Generation

In Commit Message Generation (CMG) for large commits, a model should generate a natural language description of changes performed in a single commit. The changes can be represented in different ways — in various diff formats, as separate versions of each file before and after the changes took place, and others. Moreover, models can utilize information from unchanged project files to better understand how changes impacted the project. In this work, we present a manually curated dataset for CMG tailored for larger commits.

## D.1    Dataset Structure

Each instance in the dataset represents a commit from a GitHub repository, with metadata like commit SHA and full repository name, ground truth commit message, and the list of performed changes in the Git diff format. Additionally, the dataset includes snapshots of all associated repositories to facilitate context construction. The detailed structure of each datapoint is presented in Table 12.

## D.2    Dataset Collection and Processing

We use the CommitChronicle dataset [16] as the initial source of commits for our dataset. We refer the reader to the work of Eliseeva et al. [16] for the details about data collection. In this work, we focus on Python language only and thus consider only the subset of the CommitChronicle test set that includes changes to at least one .py file.

We perform extensive filtering, including manual validation, to select high-quality examples with long diffs and commit messages. The exact data filtering steps are listed in Table 14. For the commit message quality filter, we refine the dataset released in a recent study from Li and Ahmed to make it

Table 13: The structure of a per-file modification in the commit message generation dataset.

| Field | Description |
|---|---|
| **change_type** | The type of change to the current file, one of: ADD, COPY, RENAME, DELETE, MODIFY, or UNKNOWN |
| **old_path** | The path to file before the change (might be empty if the file was created) |
| **new_path** | The path to file after change (might be empty if the file was deleted) |
| **diff** | The changes to the current file, represented in a Git diff format |

Table 14: Filters applied to the CommitChronicle subset to build the commit message generation dataset from Long Code Arena. *Since the *Quality* filter is based on a deep learning classifier, it was applied only to the subset of 3,366 commits obtained by running all the other filters.

| | Filter Description | Filter Details | Number of commits rejected by the filter (% of initial sample) |
|---|---|---|---|
| **Diff Filters** | Hash Diffs | Diff has whitespace-separated character-to-words ratio $\leq 20$ [35]. | 437 (0.25%) |
| | Modification | Diff consists only of modifications of existing files (no additions, deletions, renaming, or copying). | 25,750 (14.95%) |
| **Message Filters** | Capitalization | Message starts with an uppercase letter [42]. | 68,384 (39.70%) |
| | Verbs | Message starts with any of the curated set of verbs from the recent work of Muennighoff et al. [42]. | 90,696 (52.66%) |
| | References | Message does not contain external references (URLs or references to issues/pull requests). | 31,487 (18.28%) |
| | Noise | Message does not follow certain patterns considered automatically generated or trivial [16, 42]. | 6,304 (3.66%) |
| | Min Words | Message contains $\geq$ 4 words (whitespace-separated). | 24,474 (14.21%) |
| | Min Lines | Message contains $\geq 2$ lines. | 138,160 (80.22%) |
| | Hash Messages | Message has whitespace-separated character-to-words ratio $\leq 20$ [35] and does not contain any SHA hashes [16]. | 12,540 (7.28%) |
| | Quality | Message is considered good by the commit message quality classifier. | 106 (3.14%)* |

more suitable for data filtering purposes, and fine-tune the CodeBERT [17] model. After filtering, we retain 3,260 commits. Since we aim to target commits with larger changes, after the initial filtering, we only keep samples where the number of characters in diffs related to `.py` files is $\geq 3,000$ characters. That leaves us with 858 commits that we further filter manually. The manual labeling is conducted by one of the authors. We employ a 5-point Likert scale and additionally provide comments that elaborate on the reasoning for most of the samples. To facilitate further research, we made all the labels and comments available in the dataset.

### D.3 Evaluation

We run multiple instruction-tuned LLMs on the presented commit message generation benchmark in a zero-shot setting (*i.e.*, no examples in the prompt, only a natural language instruction). We employ the same prompt for all models. The prompt is presented in Figure 8. It was crafted through several iterations, addressing the most frequent issues in the generated messages from pilot experiments. In our main experiments, we only incorporate commit changes represented as diffs returned by the `git diff` command to prompt the LLMs. Additionally, we run the CodeT5 [59] model fine-tuned for commit message generation task on the training part of the CommitChronicle dataset. This model only takes the commit diff as an input.

We access proprietary LLMs through the official APIs. For Mixtral, Mistral, DeepSeekCoder, CodeLLaMA, and CodeT5, we use a single NVIDIA A100 GPU with default precision (except

```
Write a commit message for a given diff. Start with a heading that
serves as a summary of the whole diff: a single sentence in an
imperative form, no more than 50 characters long. If you have details
to add, do it after a blank line. Do your best to be specific, do not
use 'refactor' unless you are absolutely sure that this change is ONLY
a refactoring. Your goal is to communicate what the change does
without having to look at the source code. Do not go into low-level
details like all the changed files, do not be overly verbose. Avoid
adding any external references like issue tags, URLs or emails. Diff:

[DIFF]

Commit message:
```

Figure 8: The primary prompt for the commit message generation task.

for Mixtral, where we use 8-bit precision) and FlashAttention-2 [12] enabled. For the rest of the considered models, we use Together API.[5] For all the models, we set the temperature to 0.8 and allow them to generate up to 512 tokens. This upper bound is mostly set due to practical considerations, as the maximum length of a commit message in our dataset is only 58 whitespace-separated words. The results are presented in Table 15.

Additionally, we experiment with two alternative strategies for composing the context for the LLMs. Among the models, we select o1-mini from OpenAI as the best compromise between speed and quality among proprietary models and DeepSeek-V3, the highest-scoring OSS model in terms of ROUGE-1. We use DeepSeek-V3 tokenizer to calculate the number of tokens through the rest of the section. The first context gathering strategy is to pass the full contents of the modified files rather than diffs. Similar setting was previously employed for commit message generation by [42]. In our dataset, modified files for one commit take around 54k tokens on average, however, the maximum value is 300k, which exceeds maximum context length of 128k tokens for both o1-mini and for DeepSeek-V3. Hence, we limit the maximum allowed context length, truncating the modified files up to $\frac{max\_num\_tokens}{num\_files}$ each. We consider several upper bounds in terms of maximum context length: 4k, 8k, 16k, 32k, 64k. Due to technical limitations, we were able to obtain results for DeepSeek-V3 with contexts only up to 16k tokens. The second context gathering strategy is to further extend the prompt from our main experiments (Figure 8) with relevant context via retrieval. We use a simple BM25 [48] retriever among non-changed .py files in the corresponding repository, similar to the setting adopted by Jimenez et al. [32]. We retrieve up to 50 most relevant files and add them until the maximum context length in tokens is exceeded, possibly truncating the last file to ensure it fits the restriction on the maximum length. We consider several upper bounds in terms of maximum context length: 4k, 8k, 16k.

The results are presented in Table 16. We observe that neither of the alternative context gathering strategies leads to substantial improvements compared to our primary approach using only the commit diff. For Full File setting, the quality grows with the increase in the context size, but even at its largest (64k tokens), it remains consistently inferior to the results achieved with diffs. One reason for the inefficiency of the Full File is the large size of modifications in our dataset, which span 3.4 files on average. When including complete file contents, the input can reach up to 300k tokens. Our naive truncation strategy likely discards critical information. While additional context that facilitates better repository understanding could help generate more appropriate commit messages, BM25 retrieval might fail to uncover relevant files, leading to insignificant improvements or even degradation. Interestingly, unlike [32], we do not observe stable decrease in quality with the growth of BM25 context. We leave the exploration of more efficient and sophisticated context gathering strategies to future research.

---

[5]Together: `https://www.together.ai/`

Table 15: Results for the CMG benchmark from Long Code Arena. *R* stands for *ROUGE* metric, *BS* stands for *BERTScore* metric, where *BS (norm.)* is the normalized version. All model categories are sorted by the *ROUGE-1* metric. The best result in the category is highlighted in **bold**, and the second best result is underlined. *CodeT5 is the only model fine-tuned for the CMG task as opposed to the zero-shot setting for the rest of the models.

| | Model | BLEU | ChrF | R-1 | R-2 | R-L | BS | BS (norm.) |
|---|---|---|---|---|---|---|---|---|
| **Proprietary** | o1-preview (2024-09-12) | 4.212 | 36.38 | **29.28** | **7.66** | **20.52** | **0.8635** | **0.191** |
| | Gemini 1.5 Pro | 3.656 | 34.87 | 28.94 | 6.363 | 20.15 | 0.8593 | 0.1666 |
| | Claude 3.5 Sonnet | 4.195 | 34.85 | 28.79 | 6.134 | 19.67 | 0.8626 | 0.1857 |
| | Claude 3 Opus | **4.219** | **36.59** | 28.67 | 7.656 | 20.14 | 0.8583 | 0.1606 |
| | o1-mini (2024-09-12) | 4.09 | 34.33 | 27.96 | 6.712 | 20.05 | 0.8605 | 0.1737 |
| | Gemini 1.5 Flash | 2.918 | 34.64 | 27.38 | 5.865 | 18.68 | 0.8581 | 0.1595 |
| | GPT-4 Turbo (1106) | 2.803 | 34.39 | 26.62 | 5.296 | 17.72 | 0.8559 | 0.1462 |
| | GPT-4o (2024-11-20) | 3.066 | 34.81 | 26.07 | 5.548 | 17.65 | 0.854 | 0.1351 |
| | GPT-4o mini (2024-07-18) | 2.841 | 34.12 | 25.66 | 5.158 | 17.33 | 0.8579 | 0.1583 |
| | GPT-4 (0613) | 2.127 | 32.62 | 23.5 | 5.217 | 16.03 | 0.8522 | 0.1243 |
| | Claude 3 Haiku | 1.957 | 30.12 | 21.01 | 5.045 | 14.38 | 0.843 | 0.0695 |
| | GPT-3.5 Turbo (0613) | 2.101 | 26.664 | 19.976 | 4.227 | 14.447 | 0.846 | 0.087 |
| | GPT-3.5 Turbo (1106) | 1.885 | 20.698 | 18.424 | 3.815 | 14.087 | 0.854 | 0.136 |
| **OSS (big)** | DeepSeek-V3 (671B) | 3.788 | **35.76** | **28.63** | 6.599 | 19.81 | 0.8625 | 0.1851 |
| | Llama-3.3 (70B) | 3.751 | 33.54 | 28.38 | 6.415 | **20.12** | **0.8645** | **0.1969** |
| | Llama-3.1 (405B) | 3.563 | 34.83 | 28.25 | 6.516 | 19.94 | 0.8626 | 0.1861 |
| | Llama-3.1 (70B) | 3.634 | 34.66 | 27.62 | **6.626** | 19.27 | 0.8611 | 0.177 |
| | DeepSeek-R1 (671B) | **4.19** | 34.94 | 27.07 | 5.94 | 18.94 | 0.8644 | 0.1962 |
| **OSS (medium)** | Qwen2.5-Coder (32B) | **3.415** | **33.74** | **27.93** | **6.038** | **20.1** | **0.8616** | **0.1797** |
| | Mixtral 8 bit (8x7B) | 2.189 | 31.98 | 23.61 | 5.376 | 16.33 | 0.8476 | 0.09688 |
| | DeepSeek Coder (33B) | 1.742 | 29.08 | 21.01 | 4.471 | 14.46 | 0.8425 | 0.06697 |
| | CodeLLaMA (34B) | 1.586 | 24.632 | 17.817 | 3.684 | 13.114 | 0.844 | 0.073 |
| | QwQ (32B) | 0.529 | 14.07 | 14.66 | 3.381 | 10.26 | 0.8275 | -0.02194 |
| **OSS (small)** | Llama-3.1 (8B) | **2.409** | **31.02** | **23.66** | **4.768** | **16.67** | **0.8538** | **0.1335** |
| | Mistral (7B) | 1.895 | 30.719 | 23.648 | 4.458 | 16.262 | 0.847 | 0.096 |
| | DeepSeek Coder (6.7B) | 1.634 | 28.567 | 20.188 | 3.604 | 14.116 | 0.843 | 0.068 |
| | CodeLLaMA (13B) | 1.727 | 23.099 | 18.207 | 3.642 | 13.479 | 0.844 | 0.075 |
| | CodeLLaMA (7B) | 1.108 | 26.638 | 16.961 | 2.807 | 12.028 | 0.835 | 0.021 |
| **OSS (tiny)** | Llama-3.2 (3B) | **2.108** | **26.34** | **21.05** | **4.102** | **15.15** | **0.8461** | **0.088** |
| | DeepSeek Coder (1.3B) | 0.75 | 22.449 | 13.815 | 2.029 | 9.753 | 0.822 | -0.057 |
| | CodeT5* (220M) | 0.355 | 11.862 | 13.615 | 2.633 | 11.439 | 0.845 | 0.083 |

 # E   Bug Localization

Bug Localization task can be formulated as follows: given an issue with a bug description and a repository snapshot in a state where the bug is reproducible, identify the files within the repository that need to be modified to address the reported bug. Although this is a subset of the larger bug-fixing problem, partially covered by SWE-Bench, bug localization requires its own separate evaluation. This independent assessment can provide a better understanding of the various approaches and their efficiency in identifying the precise location of bugs within the large code bases.

## E.1   Dataset Structure

The bug localization dataset includes real issues that describe bugs, together with the respective pull requests (PRs) that fix them. Each datapoint contains three key elements: the bug description, the state of the repository where the bug is reproducible, and the list of files that need to be modified to resolve the bug. The bug description represents the body of the issue that was assigned a bug-related label. The repository state is represented by the commit SHA. The list of files that should be modified comes from the pull request that resolves the respective bug report. The full datapoint structure is presented in the Table 17

The final dataset contains 7,479 datapoints in total divided, between three sets by language:

- py — change contains only Python files (4,339 datapoints);

Table 16: Results with alternative contexts for the CMG benchmark from Long Code Arena. *R* stands for *ROUGE* metric, *BS* stands for *BERTScore* metric, where *BS (norm.)* is the normalized version. The best result for the model is highlighted in **bold**, and the second best result is underlined. The context size is reported in tokens from DeepSeek-V3 tokenizer. The context size for Diff context is the average number of tokens in diffs in our dataset.

| Model | Context Type | Context Size | BLEU | ChrF | R-1 | R-2 | R-L | BS | BS (norm.) |
|---|---|---|---|---|---|---|---|---|---|
| **o1-mini** | Diff | 2.3k | **4.09** | 34.33 | **27.96** | **6.712** | **20.05** | **0.8605** | **0.1737** |
| | Full File | 4k | 2.342 | 27.18 | 20.44 | 3.464 | 14.95 | 0.8457 | 0.0856 |
| | | 8k | 2.646 | 29.92 | 22.71 | 4.241 | 16.67 | 0.8493 | 0.1071 |
| | | 16k | 2.753 | 31.69 | 24.43 | 5.066 | 17.49 | 0.8512 | 0.1181 |
| | | 32k | 2.572 | 31.89 | 24.36 | 4.85 | 17.41 | 0.8504 | 0.1137 |
| | | 64k | 3.324 | 32.86 | 24.82 | 5.335 | 17.67 | 0.8525 | 0.1259 |
| | Diff + BM25 | 4k | 3.454 | 34.42 | 27.84 | 6.229 | 19.75 | 0.8584 | 0.1613 |
| | | 8k | 3.573 | **34.59** | 27.31 | 6.201 | 19.11 | 0.8564 | 0.1491 |
| | | 16k | 3.364 | 33.85 | 27.28 | 6.355 | 19.08 | 0.8563 | 0.1488 |
| **DeepSeek-V3** | Diff | 2.3k | **3.788** | **35.76** | 28.63 | 6.599 | 19.81 | 0.8625 | 0.1851 |
| | Full File | 4k | 2.229 | 28.88 | 21.76 | 3.507 | 15.45 | 0.8521 | 0.1237 |
| | | 8k | 2.801 | 31.34 | 24.15 | 4.81 | 17.11 | 0.8552 | 0.1421 |
| | | 16k | 3.345 | 33.59 | 26.47 | 5.647 | 18.77 | 0.859 | 0.1648 |
| | Diff + BM25 | 4k | 3.457 | 34.85 | **28.97** | **6.955** | **20.11** | **0.8631** | **0.1888** |
| | | 8k | 3.554 | 35.05 | 28.05 | 6.285 | 19.68 | 0.8627 | 0.1863 |
| | | 16k | 3.697 | 34.98 | 28.35 | 6.419 | 20.03 | 0.8627 | 0.1862 |

- `java` — change contains only Java files (2,522 datapoints);

- `kt` — change contains only Kotlin files (618 datapoints).

For each language 50 datapoints are manually verified in order to form a test subset for model evaluation (150 datapoints in total).

Based on the core fields, we calculated the number of statistics and attached them to each datapoint. The additional fields are presented in Table 18. We excluded test files from the experiment because their modifications typically only support program repairs and do not contain the actual bugs. Thus, all metrics are calculated on all project files except for the test files.

## E.2 Dataset Collection and Preprocessing

To collect the data, we use the following protocol:

1. **Collect issues, pull requests, comments.** We start with the common corpus of collected GitHub repositories. Then, for each repository, we download information about all issues, pull requests, and comments using the GitHub API. As a result, we download more than 8M issues, 7M pull requests, and 34.4M comments.

2. **Match issues with pull requests.** GitHub API does not provide information about relations between issues and pull requests. We obtain these relations by parsing references from descriptions or comments. To do so, we write regular expressions for extracting all possible referencing formats as provided in GitHub documentation. To also collect the context around the reference, we capture one "fix"-related keyword (*e.g.*, `close`, `closes`, `closed`, `fix`, `fixes`, `fixed`, `resolve`, `resolves`, `resolved`, `solve`, `solves`, `solved`) before and after the link with the regular expressions. We also check if references are mutual (if the issue refers to the pull request and vice versa) or not (if only a single link from either the issue or the pull request exists).

3. **Sort by stars.** We sort all issue-PR pairs by the number of stars in the respective repository and assign each pair an `ID` based on its index in the sorted order. We populate the `diff` field by running a git command in a locally cloned repository to get the diff in a text format. Unfortunately, this method does not work for pull requests created from forks, so we save a null value for such cases.

Table 17: Description of datapoints in the bug localization dataset.

| Field | Description |
|---|---|
| **id** | Datapoint ID |
| **repo_owner** | Bug issue repository owner |
| **repo_name** | Bug issue repository name |
| **text_id** | Datapoint text ID |
| **issue_url** | GitHub link to issue |
| **issue_title** | Issue title |
| **issue_body** | Issue body with bug description |
| **issue_labels** | List of labels assigned to issue |
| **pull_url** | GitHub link to PR |
| **pull_create_at** | Date of PR creation in format of `yyyy-mm-ddThh:mm:ssZ` |
| **base_sha** | PR base SHA |
| **head_sha** | PR head SHA |
| **diff_url** | PR diff URL between base and head SHA |
| **diff** | PR diff content |
| **changed_files** | List of changed files parsed from diff |
| **link_url** | GitHub link to issue or PR comment from which the link was parsed |
| **links_count** | Number of links between the issue and the PR, equals 2 if the link is mutual, 1 if it is one-sided |
| **link_keyword** | "Fix"-related keyword which surrounds the issue link |
| **stars** | Number of repository stars |
| **language** | Main programming language for repository |

To enhance the quality of our data, first, we apply several empirical filters and preprocessing steps based on the fields from the dataset:

1. **Select bug issues.** We retain only issues with "bug" mentioned in the labels and non-empty descriptions. Additionally, we remove issues containing links to media, as they may include crucial data visualizations that are inaccessible through other means. To ensure that most models can use the dataset for evaluation, we only keep issues written in English.

2. **Select processable changes.** For pull requests, we filter out those introducing new files and retain only pull requests modifying existing files, provided their diffs could be extracted from the cloned repository. Furthermore, to facilitate the future manual labeling process, we leave only pull requests written in Python, Java, or Kotlin, as these are languages known well to authors. To work with diffs and patches, as well as to extract the changed files and their modification modes, we use the unidiff package.[6] Additionally, we avoid pull requests that include changes to media files with non-UTF-8 encoding, as such changes are often difficult to reproduce. The most crucial filter ensures that each pull request is associated with exactly one issue, and vice versa, to maintain the relevance of changes to issue descriptions and to prevent situations where a pull request addresses multiple issues or an issue is fixed by several pull requests.

   The dataset size reduction after applying these empirical filters is summarized in Table 19. As a result of these filtering steps, 10,195 datapoints remain in the dataset.

3. **Filter outliers.** On top of the previous filtering step, we remove outliers for several numerical fields, including `changed_files_count`, `changed_lines_count`, and `issue_tokens_count`. Table 20 shows the result of removing outliers.

4. **Data analysis.** After data filtration, we are left with 7,479 datapoints that comprise the entire dataset. Table 21 presents statistics of the dataset, with the difference in statistics between languages being negligible.

5. **Manual data labelling.** After the analysis of the dataset, we carry out manual data labeling and verification process to select the subset of high-quality datapoints for evaluation. First, we sort the datapoints by the number of stars in the respective repositories, assuming that

---

[6]Undiff: `https://pypi.org/project/unidiff/`

Table 18: Description of additional metrics calculated on the bug localization dataset.

| Metric | Description |
|---|---|
| issue_symbols_count | Number of symbols in issue description |
| issue_tokens_count | Number of tokens in issue description |
| issue_words_count | Number of words in issue description |
| issue_lines_count | Number of lines in issue description |
| issue_code_blocks_count | Number of triple quotes blocks parsed in issue description |
| issue_links_count | Number of links parsed in issue description |
| diff_symbols_count | Number of symbols in diff |
| diff_tokens_count | Number of tokens in diff |
| diff_words_count | Number of words in diff |
| issue_lines_count | Number of lines in diff |
| changed_files_count | Number of all changed files mentioned in diff |
| changed_files_without_test_count | Number of changed files not including test files mentioned in diff |
| code_changed_files_count | Number of files written in Python, Java, or Kotlin mentioned in diff |
| py_changed_files_count | Number of Python files mentioned as changed in diff |
| java_changed_files_count | Number of Java files mentioned as changed in diff |
| kt_changed_files_count | Number of Kotlin files mentioned as changed in diff |
| repo_symbols_count | Total number of symbols in repository's files |
| repo_tokens_count | Total number of tokens in repository's files. |
| repo_words_count | Total number of words in repository's files |
| repo_lines_count | Total number of lines in repository's files |
| repo_files_count | Total number of files in repository |
| repo_files_without_test_count | Total number of files without tests in the repository |

popular repositories have better processes and quality for issue tracking and bug reporting. Then, we go through datapoints of each repository, selecting ones that meet the following criteria:

- The issue describes a single bug completely and exhaustively.
- The pull request is linked to the issue and resolves this issue alone.
- All changes are relevant to the described issue, with no extra functionality or side refactorings included.
- The changes were reviewed and accepted.

If a datapoint does not meet these criteria, we go to another one from the same repository, or if none are left, we move on to the next repository by the number of stars, until we select 50 good datapoints per language. To keep the distribution of the number of changed files, for each repository, we try to pick one datapoint with a single changed file and one datapoint with two or more changed files. This strategy allows us to collect a diverse set of datapoints from different repositories and keep the distribution of the number of changed files similar to the complete set of issues.

## E.3 Evaluation

We evaluate several LLMs on the bug localization task using the presented dataset.

Table 19: Empirical filters applied to the bug localization dataset.

| Field | Description | Number of data-points rejected by the filter (% of the initial set) |
|---|---|---|
| issue_labels | At least one label should include "bug" as a sub-string | 3,472,057 (79.8%) |
| issue_body | Description should not be empty | 16,265 (0.37%) |
| issue_body | Description should contain only text without attached media | 145,225 (3.34%) |
| issue_body | Description should be written mostly in English | 35,942 (0.83%) |
| diff | Diff can be extracted and should not be empty or corrupted | 475,447 (10.93%) |
| diff | Diff should consist only of modifications of existing files and no introduction of new files | 30,572 (0.7%) |
| diff | Diff should include at least one file in either Python, Java, or Kotlin | 138,653 (3.19%) |
| diff | Diff should include only UTF-8 files to filter out unreadable or graphical objects | 18 ($\leq 0.01\%$) |
| base_commit | Repository content on base commit can be extracted and should not be empty or corrupted | 6,198 (0.14%) |
| pull_url | PR should refer to no more than one issue | 7,376 (0.17%) |
| issue_url | Issue should refer to no more than one pull request | 1,934 (0.04%) |
| link_keyword | "fix"-related keyword should stay before or after link in the issue description. | 10,406 (0.24%) |

Table 20: Outlier filters applied to the bug localization dataset.

| Field | Description | Number of data-points rejected by the filter (% of initial set) |
|---|---|---|
| changed_files_count | Number of changed files should not be more than 22 (0.99 quantile) | 100 ($\leq 0.01\%$) |
| changed_lines_count | Number of changed lines should not be more than 594 (0.99 quantile) | 102 ($\leq 0.01\%$) |
| issue_tokens_count | Issue description can be tokenized using GPT-4 tokenizer | 43 ($\leq 0.01\%$) |
| issue_tokens_count | Issue description should contain at least 13 tokens (0.01 quantile) | 85 ($\leq 0.01\%$) |
| issue_tokens_count | Issue description should contain no more than 4,500 tokens (0.99 quantile) | 103 ($\leq 0.01\%$) |

For all models, we adopted a unified prompt structure (Figure 9), which includes the repository name, issue title, and description, along with optional additional context.

First, we evaluate two context-filling strategies to understand how context influences the quality of bug localization and how it can be optimized for more efficient use by LLMs in solving this particular task:

- *Only issue description context.* This configuration only considers the issue description as context to determine whether it contains sufficient information for bug localization. It also serves to analyze the potential impact of data contamination.

- *Repo file paths list.* This strategy adds a list of all files in the repository as context, enabling the model to utilize structural information from the codebase. This approach assesses

Table 21: Final statistics of the dataset.

| Field | Min | Median | Mean | Max |
|---|---|---|---|---|
| **repo_files_count** | 16 | 331 | 1,077 | 33,644 |
| **repo_lines_count** | 9 | 52,743 | 145,377 | 8,687,912 |
| **repo_tokens_count** | 78 | 488,286 | 1,684,619 | 225,649,725 |
| **changed_files_count** | 1 | 1 | 2 | 21 |
| **changed_lines_count** | 1 | 15 | 37 | 594 |
| **changed_tokens_count** | 1 | 158 | 608 | 837,626 |
| **issue_words_count** | 1 | 106 | 149 | 1,806 |
| **issue_lines_count** | 1 | 22 | 33 | 586 |
| **issue_tokens_count** | 13 | 227 | 432 | 4,491 |
| **issue_links_count** | 0 | 0 | 0.80 | 56 |
| **issue_code_blocks_count** | 0 | 1 | 0.99 | 31 |

```
SYSTEM:
You are an AI assistant specialized in software bug localization.
Your task is to identify the MOST likely files to be modified to
fix the given bug. You will be provided with the repository name and
a GitHub bug issue description*. Analyze the issue description and
determine the files in the repository that are MOST likely to require
modification to resolve the issue. Provide the output in JSON format
with the list of file paths under the key "files".
Provide JSON ONLY without any additional comments.

USER:
GitHub repo name:
[REPO_OWNER/REPO_NAME]

Issue description:
[ISSUE_TITLE]
[ISSUE_BODY]

[CONTEXT]
```

Figure 9: Prompt for bug localization. *Can slightly vary to describe the content and structure of the context provided.

whether the mere presence of file names aids effective bug localization. To prioritize the most relevant file paths in the context, we employed the following algorithm:

1. **Ranking.** We use a simple NLTK tokenizer and BM25 to rank the files in the repository based on their lexical similarity to the issue description.
2. **Filling.** Based on the ranking, we concatenate the context for each file (file path along with imports).
3. **Cutting.** Since the context appears last in the prompt, we trim the final message to fit the total context size of each model.

The expected output of the LLMs is a list of files which contain bugs. To measure the quality of this output and compare it with the expected list of buggy files, we calculate the following metrics:

- **P (Precision).** This metric shows how many predicted files were correct.
- **R (Recall).** This metric shows how many actual bugged files were correct.
- **FPR (False Positive Rate).** This metric shows how many non-buggy files were incorrectly predicted.
- **F1-score**. The balance between Precision and Recall.

Table 22: The baseline results for the bug localization task without additional context.

| Model | Context Size | P | R | F1-score | FPR | All correct | All incorrect | # Output |
|---|---|---|---|---|---|---|---|---|
| o1 | 128k | 0.299 | 0.286 | 0.255 | 0.015 | 0.07 | 0.55 | 1.97 |
| GPT-4o | 128k | **0.303** | **0.305** | **0.270** | **0.018** | **0.12** | **0.54** | **2.29** |
| GPT-4o mini | 128k | 0.112 | 0.164 | 0.117 | 0.042 | 0.03 | 0.77 | 3.79 |
| GPT-3.5 Turbo (1106) | 16k | 0.219 | 0.178 | 0.177 | 0.017 | 0.09 | 0.73 | 1.93 |
| Gemini 1.5 Pro | 1M | **0.309** | **0.294** | **0.270** | **0.020** | **0.14** | **0.55** | **2.52** |
| Claude 3 Opus | 200k | - | - | - | - | - | - | - |
| Claude 3 Haiku | 200k | - | - | - | - | - | - | - |
| Claude 3.5 Sonnet | 200k | 0.199 | 0.254 | 0.196 | 0.021 | 0.05 | 0.61 | 3.16 |
| Claude 3.5 Haiku | 200k | 0.212 | 0.256 | 0.211 | 0.026 | 0.08 | 0.61 | 2.76 |
| Llama-3.2 (3B) | 128k | 0.114 | 0.215 | 0.130 | 0.158 | 0.0 | 0.74 | 3.11 |
| Llama-3.1 (8B) | 128k | 0.072 | 0.143 | 0.084 | 0.056 | 0.01 | 0.81 | 5.60 |
| Llama-3.1 (70B) | 128k | 0.156 | 0.196 | 0.157 | 0.035 | 0.05 | 0.72 | 3.90 |
| Llama-3.1 (405B) | 128k | - | - | - | - | - | - | - |
| Qwen2.5 (7B) | 128k | 0.172 | 0.141 | 0.140 | 0.016 | 0.08 | 0.79 | 2.00 |
| Qwen2 (72B) | 128k | 0.191 | 0.157 | 0.159 | 0.023 | 0.09 | 0.76 | 2.45 |
| DeepSeek R1 (671B) | - | - | - | - | - | - | - | - |
| DeepSeek V3 (671B) | - | - | - | - | - | - | - | - |

- **All correct.** The percentage of cases where all files were correctly identified.

- **All incorrect.** The percentage of cases where all files were incorrectly identified.

- **# Output.** The average number of buggy files detected.

All results are presented in two separate tables: Table 22 reports results for the small-context setting, while Table 23 presents results for the large-context setting. The evaluation demonstrated that even a simple additional context can double the effectiveness of bug localization. In small-context settings, the average token usage is less than $1k$ (minimum: $149$, maximum: $149$), whereas, in large-context settings, it reaches approximately $10k$ (minimum: $251$, maximum: $> 200{,}000$). This indicates that, for certain data points, even larger contexts can be provided, potentially leading to higher scores.

However, we observed an interesting pattern in LLaMA-based models: increasing the context size adversely affected their performance. Specifically, with larger contexts, these models often produced excessively long lists of files or failed to generate JSON outputs in the correct format. This mean that the context and the output format should be kind of model specific and not universal. This suggests that both context handling and output formatting are model-specific rather than universally applicable.

# F  Module Summarization

For the Module Summarization task, the model should write textual documentation based on the module's or project's source code and intent (a one-sentence description of the expected documentation content). This task greatly increases the context size available to the models compared to the existing benchmarks that cover method- or class-level summarization.

## F.1  Dataset Collection and Processing

The dataset consists of the datapoints with their structure as in Table 24.

To collect the data, we use the following protocol:

1. We start with the Python subset of the common corpus of GitHub repositories. For each repository, we extract documentation files — files with extensions `.md`, `.txt`, and `.rst`, located in the `docs` directory of the repository.

Table 23: The baseline results for the bug localization task with file paths list context.

| Model | Context Size | P | R | F1-score | FPR | All correct | All incorrect | # Output |
|---|---|---|---|---|---|---|---|---|
| o1 | 128k | **0.622** | **0.630** | **0.576** | **0.010** | **0.28** | **0.15** | **2.22** |
| GPT-4o | 128k | 0.535 | 0.635 | 0.527 | 0.012 | 0.23 | 0.12 | 2.85 |
| GPT-4o mini | 128k | 0.350 | 0.666 | 0.416 | 0.035 | 0.07 | 0.13 | 5.44 |
| GPT-3.5 Turbo (1106) | 16k | 0.436 | 0.497 | 0.421 | 0.021 | 0.17 | 0.31 | 3.35 |
| Gemini 1.5 Pro | 1M | 0.471 | 0.671 | 0.501 | 0.015 | 0.17 | **0.09** | 3.55 |
| Claude 3 Opus | 200k | 0.471 | 0.637 | 0.481 | 0.018 | 0.2 | 0.1 | 3.77 |
| Claude 3 Haiku | 200k | 0.429 | 0.59 | 0.441 | 0.029 | 0.13 | 0.2 | 4.04 |
| Claude 3.5 Sonnet | 200k | 0.461 | 0.748 | 0.523 | 0.017 | 0.13 | 0.11 | 3.48 |
| Claude 3.5 Haiku | 200k | 0.553 | 0.741 | **0.583** | 0.038 | 0.22 | 0.1 | 2.88 |
| Llama-3.2 (3B) | 128k | 0.268 | 0.748 | 0.321 | 0.204 | 0.14 | 0.1 | 18.10 |
| Llama-3.1 (8B) | 128k | 0.234 | 0.737 | 0.305 | 0.145 | 0.05 | 0.1 | 16.03 |
| Llama-3.1 (70B) | 128k | 0.287 | 0.664 | 0.351 | 0.041 | 0.05 | 0.13 | 8.37 |
| Llama-3.1 (405B) | 128k | 0.432 | 0.639 | 0.465 | 0.025 | 0.16 | 0.14 | 4.36 |
| Qwen2.5 (7B) | 128k | 0.559 | 0.572 | 0.517 | 0.013 | 0.25 | 0.22 | 2.79 |
| Qwen2 (72B) | 128k | 0.431 | 0.686 | 0.483 | 0.026 | 0.14 | 0.1 | 5.16 |
| DeepSeek R1 (671B) | 128k | 0.529 | 0.68 | 0.538 | 0.021 | 0.2 | 0.1 | 3.04 |
| DeepSeek V3 (671B) | 128k | 0.489 | 0.697 | 0.523 | 0.025 | 0.19 | **0.08** | 3.61 |

Table 24: The structure of datapoints in the module summarization dataset.

| Field | Description |
|---|---|
| **repo** | The full name of the GitHub repository the commit comes from |
| **docfile_name** | The name of the documentation file. May be useful in the prompt |
| **intent** | Small manually gathered intent that describes what we expect from the generated documentation |
| **license** | The type of the license in the repository of the commit |
| **path_to_docfile** | The path to file with documentation in the repository |
| **relevant_code_files** | List of paths in the repository to the potentially relevant code files |
| **relevant_code_dir** | Directory with relevant code, field can be empty |
| **target_text** | The text of the target documentation — ground truth in our task |
| **relevant_code_context** | Code context joined from relevant code files and directories |

2. For each documentation file, we extract the associated source code. To do this, we parse the target documentation and extract names of all code files and directories mentioned in it. If a file does not contain any such mentions, we skip it.

3. To further filter the documentation files, we convert documentation into a plain text format by removing specific Markdown syntax (as well as text between Markdown tags like *code*, *autosummary*, etc.). We then ensure that each document contains valuable information and has at least 10 lines of text remaining after cleaning. Since the filtering is quite strict, we believe that only important documents remain after this stage.

4. We perform manual review of the datapoints to ensure that the content contains not only information about the code but also summarizes the entire module or project. After manual review, we leave 216 out of 461 files. Most of the files that we reject contain non-informative

text that is not related to code. Also, for each documentation file, we manually specify an intent that the model under evaluation can use during generation.

- Manual verification is essential, as our experience with data frequently reveals instances where a docfile lacks useful content or does not provide substantial information in the plain text format, without special extensions that enrich documentation.

## F.2 Evaluation

- We run several LLMs on the collected module summarization dataset with different length of the relevant code context. To assess the quality of the generated documentation, we introduce a new metric called CompScore that uses LLM (Mistral-7B in our case) as an assessor. CompScore feeds the assessor LLM relevant code and two versions of documentation: the ground truth and the model-generated text. The LLM then evaluates which documentation better explains and fits the code. To mitigate variance and potential ordering effects in model responses, we calculate the probability that the generated documentation is superior by averaging the results of two queries:

$$\text{CompScore} = \frac{P(\text{pred} \mid \text{LLM}(\text{code}, \text{pred}, \text{gold})) + P(\text{pred} \mid \text{LLM}(\text{code}, \text{gold}, \text{pred}))}{2}$$

To count $P(\text{pred} \mid \text{LLM}(\text{code}, \text{pred}, \text{gold}))$, we follow several steps:

1. Construct the prompt and feed it into the assessor LLM (see Figure 10).

```
I have 2 different documentations about {intent}. Decide which
documentation is better: documentation A or documentation B.
My code: [TRIMMED_CODE_CONTEXT]
Documentation A: [PREDICTED_DOC]
Documentation B: [GROUND_TRUTH_DOC]
Better documentation is documentation
```

Figure 10: Prompt for the CompScore metric.

2. Get logits for the next token being "A" and "B" ($logit_A$ and $logit_B$) and convert them into probabilities:

$$prob_A, prob_B = \exp\left(log\_softmax([logit_A, logit_B])\right)$$

3. $P(\text{pred} \mid \text{LLM}(\text{code}, \text{pred}, \text{gold})) = prob_A$ shows the probabilty that the predicted documentation is better than the original from the perspective of the LLM assessor.

- For our experiments, we use Mistral-7B-Instruct-v0.2 as LLM assessor. We truncate relevant code up to 6,000 tokens in the prompt for metric computation. We evaluate all the models presented in Table 25 via OpenAI API or TogetherAI API with the same generation parameters. We use zero temperature and predict up to 2,000 new tokens without any penalties to get deterministic results during generation. Table 26 shows the results for all the evaluated LLMs with varying length of available relevant code context.

- We observe that both increasing the context size and the size of the model leads to higher quality. The o1 model outperforms the others, achieving a notable CompScore of 72.22. Interestingly, the CodeLlama and Llama3 models show worse performance than the Llama2 model on small contexts. Although doubling the context size does not significantly impact the CompScore, a substantial difference emerges when comparing the metrics for the smallest and largest context windows. Investigating which context is most relevant for this task, as well as exploring different context composition strategies, is left for future research.

Table 25: CompScore metric in the module summarization benchmark for various LLMs.

| Model | 128 tokens | 512 tokens | 1k tokens | 2k tokens |
|---|---|---|---|---|
| Mistral-7B-v0.3 | 35.84 | 39.18 | 41.03 | 46.23 |
| Mixtral-8x7B | 34.63 | 38.48 | 39.96 | 40.89 |
| Mixtral-8x22B | 35.33 | 38.48 | 39.49 | 42.24 |
| Llama2-7B | 36.33 | 44.21 | 44.13 | 46.19 |
| Llama2-13B | 40.96 | 47.37 | 46.57 | 48.12 |
| Llama2-70B | 39.78 | 45.97 | 46.37 | 48.24 |
| CodeLlama-7B | 33.02 | 36.88 | 36.49 | 38.06 |
| CodeLlama-70B | 38.36 | 38.74 | 39.76 | 37.23 |
| Llama3-8B | 25.37 | 32.14 | 33.84 | 37.35 |
| Llama3-70B | 24.79 | 30.08 | 33.18 | 36.45 |
| Gemma-2B | 16.43 | 21.04 | 21.85 | 25.38 |
| Gemma-7B | 24.16 | 28.24 | 30.44 | 33.96 |
| GPT-3.5 | 36.83 | 41.59 | 45.59 | 49.48 |
| GPT-4 | 45.62 | 52.59 | 56.22 | 57.33 |
| o1 | **63.53** | **63.99** | **65.10** | **66.33** |
| gpt-4o | 58.27 | 61.67 | 63.74 | 65.95 |
| Llama3.3-70B-Instruct | 51.03 | 54.30 | 56.49 | 59.67 |
| Qwen2.5-72B-Instruct | 59.27 | 63.15 | 65.14 | 66.37 |
| deepseek-ai-DeepSeek-V3 | 59.27 | 63.15 | 65.14 | 66.37 |
| deepseek-ai-DeepSeek-R1 | 61.53 | 62.49 | 64.20 | 64.87 |

Table 26: CompScore metric in the module summarization benchmark for various LLMs on large contexts.

| Model | 4k tokens | 8k tokens | 16k tokens | 64k tokens | 100k tokens |
|---|---|---|---|---|---|
| o1 | **68.36** | **69.93** | **70.93** | **71.53** | **72.22** |
| gpt-4o | 66.61 | 66.96 | 67.02 | 68.09 | 68.12 |
| Llama3.3-Instruct | 60.54 | 61.3 | 62.86 | 63.14 | 64.20 |
| Qwen2.5-72B-Instruct | 67.72 | 68.44 | 68.73 | 69.25 | 69.73 |
| deepseek-ai-DeepSeek-R1 | 66.51 | 67.45 | 66.62 | - | - |

