# OpenReview forum: "Long Code Arena: a Set of Benchmarks for Long-Context Code Models"
_NeurIPS.cc/2025/Datasets_and_Benchmarks_Track — Submitted to NeurIPS 2025 Datasets and Benchmarks Track_

### Official Review · Reviewer_2hv5 · 2025-06-17

**Rating:** 4
**Confidence:** 4

**Summary:**

This paper presents Long Code Arena, a suite of benchmarks for ML4SE models covering six tasks. The tasks are designed to require models to use project information without complex interactions. Data is sourced from open-source repositories with permissive licenses, and baseline solutions are provided for each task. The paper aims to guide future research, focusing on improving task approaches and data collection strategies.

**Dataset Code Accessibility:**

Yes

**Ethical Considerations:**

No, there are no or only very minor ethics concerns

**Final Justification:**

The author has not addressed the new concerns raised, particularly regarding task definition and the details of dataset setup. Therefore, I keep my score.

**Limitations Weaknesses:**

1. The statement that code completion requires a base model rather than an instruction-tuned model is inappropriate; an instruction-tuned model can also be used for code completion. The purpose of selecting only two tasks is unclear. The tasks should be chosen based on the output, such as code and text tasks.

2. The definition of the library-based code generation task as generating a single file is inaccurate; the goal should be to generate a function or fulfill a specific requirement. Evaluating only the recall rate of APIs in context does not reflect the correctness of the generated code; it seems more like an API retrieval or recommendation task rather than a code generation task.

3. The evaluation metric EM for the repository completion task should consider leading and trailing spaces for Python, which is indentation-sensitive.

4. I could not find a specific description of “file-level’’, nor how context is formed in the repository completion task. Especially for baselines that are not fine-tuned with instructions, does the context include file paths or just code snippets?

5. The difference in the "Contextl" column of Table 3 is not explained. What does it specifically refer to?

**Strengths Contributions:**

1. The dataset of CI build Repair provides run-based evaluation to assess correctness.

2. The dataset of Project-Level code completion uses history from Git to mimic the real-world use case and avoid possible data leakages.

3. Long Code Arena is a suite of novel benchmarks for ML4SE models that cover six different tasks.

---

> ### Author Rebuttal · Authors · 2025-07-31
>
> We deeply thank the reviewer for constructive and detailed feedback. We address the raised points as follows.
> 1. Indeed, instruction-tuned models can be used for the completion task. However, they may face latency issues in practice, as it is easier to continuously serve queries from plain autocompleting models with no instruction formatting as user types in the IDE. For example, models used in JetBrains IDEs are tuned for completion rather than instruction following [1]. Choosing the two tasks for presentation in the main text based on the output type is a great suggestion. Nevertheless, we hope that detailed descriptions for all the tasks in the Appendix will also prove to be useful for the readers.
> 2. The task in its current formulation is derived from file-level examples of usage for a given library, which may span many functions and dozens to hundreds lines of code, that’s why it is file-level. While we do not report it in the current version of the paper, we also compute additional metrics for all tasks and report them in the leaderboard. For code generation they include ChrF, which serves as a proxy for usefulness and correctness of the generated code, as shown in previous studies [2]. We will update the paper to include thorough descriptions of all the metrics that we compute and report the models’ scores according to them.
> 3. We agree that leading spaces are essential for Python syntax. However, we decided to report this particular metric, because it is robust to possible variations in indentation conventions (tabs vs whitespaces) and in most editors developers have the indentation set up correctly when moving to the next line. Also, Exact Match with trimmed whitespaces is an upper bound for the EM that takes indentation into account, and our experiments show very slight differences between various EM versions.
> 4. File-level context composer is a composer that always returns an empty string as a context, i.e., the input sequence is a file prefix only, with no repository-level data. For all the presented models we follow the input sequence patterns from their original papers/reports. Typically, file paths are included and separated by special tokens.
> 5. Difference is the EM gain when empty context is compared with best-perplexity context. We will  update the table so everything reads clearer.
>
> [1]  Introducing Mellum: JetBrains’ New LLM Built for Developers. JetBrains blog. https://blog.jetbrains.com/blog/2024/10/22/introducing-mellum-jetbrains-new-llm-built-for-developers/
>
> [2] Evtikhiev, Mikhail, et al. "Out of the bleu: how should we assess quality of the code generation models?." Journal of Systems and Software 203 (2023): 111741.

---

> > ### Comment · Reviewer_2hv5 · 2025-08-06
> > **Response**
> >
> > Thank you for your rebuttal. After reading the rebuttal:
> > 1. I still believe that the statement claiming code completion requires a base model is incorrect. Instructions can be automatically completed through system presets without the need for users to manually fill them in.
> > 2. I emphasize the importance of task definition: the goal of a task should be specific and meaningful, rather than broadly defined. Moreover, hundreds of lines of code are evaluated using traditional metrics rather than executable ones, which cannot guarantee the accuracy of the evaluation and weaken the usability of the dataset.
> > 3. When there is a large number of nested $if$ and $else$ statements in code, different indentations convey different meanings. It is not possible to simply rely on automatic line breaks. The lack of precise EM judgment may lead to misjudging certain situations. The scarcity of such data in the dataset should not be used as a reason to overlook this issue.

---

### Official Review · Reviewer_vApX · 2025-06-30

**Rating:** 4
**Confidence:** 4

**Summary:**

This paper presents Long Code Arena, a benchmark suite designed to evaluate the performance of code-oriented LLMs on tasks that require project-level or module-level context. It includes six diverse tasks:
1) Library-based Code Generation: Generate full scripts using a given library’s API, evaluated via API Recall.
2) CI Build Fixing: Generate patches to fix real GitHub Actions failures, evaluated through actual build results.
3) Project-level Code Completion: Complete single-line code using project history as context, evaluated by exact match and perplexity.
4) Commit Message Generation: Generate natural language summaries for large diffs, evaluated using BLEU/ROUGE/BERTScore.
5) Bug Localization: Predict which files need to be changed given a bug description and code snapshot, evaluated with precision/recall/F1.
6) Module Summarization: Generate high-level documentation for modules, evaluated using a novel LLM-based CompScore metric.

For each task, the authors curate datasets from licensed GitHub repositories, apply task-specific filtering and manual validation, define clear evaluation metrics, and provide open-source baselines across a range of models.
The paper evaluates 20 LLMs and finds significant performance gaps—especially in long-context scenarios—highlighting that these challenges remain largely unsolved.

**Dataset Code Accessibility:**

Yes

**Ethical Considerations:**

No, there are no or only very minor ethics concerns

**Final Justification:**

After the rebuttal, all my concerns have been addressed, and I have no further questions. I will maintain my boarding accept score.

**Limitations Weaknesses:**

1. The benchmark currently focuses primarily on Python, with limited coverage of other important programming languages (e.g., Java, Kotlin for bug localization). This leaves major ecosystems like JavaScript and C++ untested, limiting generalizability.

2. Since the data is collected from popular open-source repositories, there is a risk of overlap with LLM pretraining corpora. While using historical commits helps mitigate this issue, the validity of the benchmark could be strengthened with more aggressive data decontamination or a strict unseen-project split.

**Strengths Contributions:**

1. The benchmark targets a highly relevant and practical area, and the release of code, datasets, and models provides substantial value to the community.

2. This paper conduct detailed evaluations across 20+ LLMs, offering deep insights through extensive empirical analysis.

3. The benchmark design spans a wide range of realistic tasks, covering both generative and retrieval-style objectives, and includes novel evaluation metrics (e.g., CompScore).

---

> ### Author Rebuttal · Authors · 2025-07-31
>
> We deeply thank the reviewer for constructive and detailed feedback. We address the raised points as follows.
> 1. Our study focuses on Python because it is the dominant language in the target domain and widely used in prior work to ensure fair comparisons. We evaluate across six diverse benchmarks and aimed to provide comprehensive coverage within this scope. Having said that, we are actively maintaining the benchmark and have already extended some tasks to other languages, including Kotlin and Java. We will add this to the revised manuscript.
> 2. We acknowledge the reviewer’s concern regarding potential data overlap with LLM pretraining corpora. While proprietary models’ training data is indeed unknown, we mitigated memorization risks by using data that should not appear in the pretraining corpora in the exact same form as in our datasets. It follows from using data from historic examples, newly crafted instructions, distant parts of repositories, all of which require targeted efforts to include them. As an additional sign of limited memorization, large proprietary LLMs still struggle to show high scores in our benchmarks. We will extend our discussion of memorization in Section 5 in the revised manuscript.

---

### Official Review · Reviewer_RLq7 · 2025-07-01

**Rating:** 4
**Confidence:** 3

**Summary:**

In this paper, the authors present Long Code Arena, a comprehensive benchmark suite for evaluating code models for long-context software engineering tasks. It includes six benchmarks covering Library-Based Code Generation, Project-Level Code Completion, CI Build Repair, Commit Message Generation, Bug Localization, and Module Summarization. The datasets are sourced from open-source repositories, carefully filtered, and manually reviewed. Each benchmark is accompanied by metrics, benchmarks, and evaluation suites. The authors emphasize realism, long-term dependencies, and task diversity, thus filling a clear gap in current ML4SE benchmarks. The paper presents the evaluation of datasets in detail and leaves the evaluation of other datasets for the appendices.

**Additional Feedback:**

- The authors chose to present two tasks in more detail. Since the purpose of the article was to present the new benchmark, it would have been preferable to present the different tasks in detail and keep some evaluations in the appendices.
- The discussions and comparisons with the state of the art could be a little more detailed to better help the reader understand the work.
- For the evaluation of code generation, the metric measuring the number of API calls may seem insufficient. How can we know if the calls are made in the right place and make sense?
- For the evaluation of code completion, the metric measuring exact matches seems a bit strict. What happens if the code is correct but not exactly the same?
- In Section 3, the choice to consider scores below 10% as identical is a bit surprising. What is the justification? To reduce the effect of randomness, repeating the experiments several times could also be a solution.
- In Table 4, it is not clear where the results for LB-CG come from, since they are different from those presented in Table 1. What is the explanation?

**Dataset Code Accessibility:**

Yes

**Dataset Code Comments:**

Data and codes are available. There are sufficient details for reproducibility.

**Ethical Considerations:**

No, there are no or only very minor ethics concerns

**Final Justification:**

A clear and well-structured response to my review has been provided by the authors.
All of my comments have been addressed with extensive and precise details.
The final paper has been revised accordingly.
With these clarifications and modifications, my evaluation has been raised from 4 to 5.

**Limitations Weaknesses:**

- The benchmarks primarily target the Python language. The lack of evaluation on other languages may limit generalizability.
- Since many benchmarks are derived from open-source code, the models may have been trained on overlapping data. Although the authors acknowledge this issue and mitigate it through timestamp-based filtering, it may merit further empirical analysis.
- For the code generation task, manually generating instructions via GPT-4 can introduce subtle biases (even if they are edited by humans).
- Using a single metric to evaluate different tasks seems unlikely to ensure robust evaluation and comparison.

**Strengths Contributions:**

- The paper is clear and well-organized, making it easy to understand.
- The proposed benchmark is very substantial and benefits the community with data coming from numerous Github repositories.
- Several tasks are proposed and reflect real-life developer workflows: continuous integration repair, CMG for large commits, bug tracking, etc.
- A wide range of popular models are tested with long-context and zero-shot variants.

---

> ### Author Rebuttal · Authors · 2025-07-31
>
> We deeply thank the reviewer for constructive and detailed feedback. We address the raised points as follows.
> 1. Our study focuses on Python because it is the dominant language in the target domain and widely used in prior work to ensure fair comparisons. We evaluate across six diverse benchmarks and aimed to provide comprehensive coverage within this scope. Having said that, we are actively maintaining the benchmark and have already extended some tasks to other languages, including Kotlin and Java. We will add this to the revised manuscript.
> 2. We will discuss it further in Section 5, as per our comment to reviewer QfsZ.
> 3. That is a fair point, we will report it more explicitly in Section 2.1 to ensure that readers are well-informed.
> 4. Thanks for pointing it out! While we have a single “main” metric for each task, we also compute “supporting” metrics and report them in the HuggingFace leaderboard. Unfortunately, this part is not fully represented in the current version of the manuscript due to the space limitations. For example, for library-based code generation we compute ChrF with respect to the reference solution as a proxy for generation usefulness, following the previous work on metrics for code generation [1]. We will extend the paper with (i) descriptions of all metrics that we compute and (ii) tables with results for supporting metrics, to ensure readers can see the full picture of what is available for the benchmarks.
>
> ### Additional Feedback
> 1. Before sticking to the current paper structure with two tasks described in details in the main part of the paper, we have tried to provide technical descriptions of all six tasks in the main text and move information on experiments to Appendix. Unfortunately, such structure leads to omitting a lot of important details due to the space limitations. Thus, we made a deliberate choice to organize the paper in its current form, which, we agree, has its downsides.
> 2. Thanks, we will further update them, as per our comment to reviewer QfsZ.
> 3. As described above, we use supporting metrics alongside the API Recall in order to serve as a proxy of correctness and usefulness. We hope that it answers the question, and we will include further explanation of it into the paper.
> 4.  EM is strict indeed. However, as was shown in [1] it is still a valid proxy metric. We agree with the reviewer that there are possible flaws of EM in that particular setting when the code is the same but is written slightly differently. Yet, we argue that for the enterprise use case, following the project-level conventions is also important. Moreover, we evaluated a few different metrics (edit similarity, ChrF) during the paper preparation, and all of them were highly correlated. If the reviewer finds that useful, we can include these results in the Appendix.
> 5. We will replace the usage of the threshold with analysis based on bootstrapping for paired data.
> 6. Thanks for pointing it out. Unfolding the differences:
>   (i) for o1, that’s the difference in rounding (0.45 / 0.44), it will be consistent 0.44 everywhere in the next paper version
>   (ii) For Claude-3.5-Haiku, results from Claude-3-Haiku sneaked into Table 4. Will be replace them with the correct value of 0.40
>   (iii) Gemini-1.5 Pro is reported in Table 4 and HuggingFace leaderboard, but was omitted in Table 1, we will add it in table 4
>
> [1] Evtikhiev, Mikhail, et al. "Out of the bleu: how should we assess quality of the code generation models?." Journal of Systems and Software 203 (2023): 111741.

---

### Official Review · Reviewer_QfsZ · 2025-07-09

**Rating:** 5
**Confidence:** 4

**Summary:**

The paper presents long-code-arena, a suite of six benchmarks for code processing tasks that require project-level context. These tasks cover different code processing types that include library-based code generation, bug localization, and module summarization. The paper presents a manually verified dataset for testing, an evaluation suite, and open-source baseline solutions based on popular LLMs to demonstrate the usage of the dataset. The benchmark is published on HuggingFace spaces with a leaderboard.

**Additional Feedback:**

1. In Figure 1, is comma "," used between values used by mistake? I assume they indicate decimal point?

**Dataset Code Accessibility:**

Yes

**Ethical Considerations:**

No, there are no or only very minor ethics concerns

**Final Justification:**

I am sticking to my score as the authors mentioned in the rebuttal that they will have the related discussions in the camera ready.

**Limitations Weaknesses:**

I do not have strong points to raise as weaknesses. However, there are a few minor points I can quickly raise:

1. There are benchmarks at project-level code generation. Authors have cited RepoEval and RepoBench, but they should cover other works too. Also, seems like authors mis-represent the literature, e.g., RepoEval consists of a function completion dataset (it is not limited to single line code completion). It would be great if long-code-arena benchmarks are compared with existing datasets. A table to show the differences would be helpful. Authors should note that this work is not the first work which attempts to construct such dataset, though the paper is trying to establish a common platform for long-code-understanding evaluation of LLMs and maintain a leaderboard to consolidate the research progress in this direction.

2. The authors argued that the tasks chosen in this work are less prone to model memorization and I find their reasoning a bit weak, specially when we do not know many details of the proprietary models.

**Strengths Contributions:**

- Long Code Arena addresses evaluation of LLMs for code processing tasks that go beyond a single of context, enabling evaluation of long context understanding of the models. Notably, this is an extremely important aspect of LLM evaluation.
-  The paper is clearly written, I liked the idea of decomposing section 2 into 3 parts, focusing on the target tasks.
- The dataset collection details are provided adequately. This will help future works to develop newer datasets.
- The dataset is constructed by respecting licenses, demonstrating responsible use of existing data at GitHub.
- The choice of experiment models are quite comprehensive (e.g., Table 1). I would prefer choosing Qwen2.5-Coder-32B over code-llama-7B for project-level code completion and other tasks.
- The paper discusses the limitations of the work that demonstrates authors' awareness and acknowledgement of the limitations.

---

> ### Author Rebuttal · Authors · 2025-07-31
>
> We deeply thank the reviewer for constructive and detailed feedback. We address the raised points as follows.
> 1. We will extend the review of related works, especially in the part related to repository-level code generation. For RepoEval, the original work introduces datasets for three tasks: line completion, API invocation, and function completion. Our paper primarily focuses on line completion, but we agree that including a detailed comparison with the other two tasks will strengthen the discussion. In contrast to function completion in RepoEval, we do not assess functional correctness but rather measure similarity to the reference code snippet and identify which reference library-specific APIs were called in the generated code fragment. On the other hand, it allows us to use long snippets of code that go beyond the scope of a single function.
> 2. We acknowledge the reviewer’s concern regarding potential data overlap with LLM pretraining corpora. While proprietary models’ training data is indeed unknown, we mitigated memorization risks by using data that should not appear in the pretraining corpora in the exact same form as in our datasets. It follows from using data from historic examples, newly crafted instructions, distant parts of repositories, all of which require targeted efforts to include them. As an additional sign of limited memorization, large proprietary LLMs still struggle to show high scores in our benchmarks. We will extend our discussion of memorization in Section 5 in the revised manuscript.
> 3. Thanks for noticing the typo in Figure 1, where a comma was mistakenly used instead of a decimal point. This will be corrected in the revised manuscript.

---

### Note · Authors · 2025-08-15

We deeply thank the reviewers for their constructive and detailed feedback. We will update the manuscript based on this feedback, as stated in our rebuttals, to make the paper easier to follow, fix typos, and extend the review of related work. In addition, following the response of reviewer 2hv5, we will update the default metric used in the code completion benchmark to include whitespaces. Please note that, based on our evaluation, the results after updating the metric change only marginally and do not affect any of the conclusions or claims we make.

---

### Decision · Program_Chairs · 2025-09-18

**Decision:**

Reject

**Comment:**

The paper introduces Long Code Arena, a comprehensive suite of six benchmarks related to code generation, repair, completion, and summarization - addressing a critical gap in the field in evaluating long context models. All the reviewers agree that the paper addresses a very important problem. The authors did an excellent job in providing a detailed rebuttal that addressed all key weaknesses/concerns about metrics and evaluation details. Overall, I believe that this new benchmark is a significant and very useful resource for the community, and therefore I recommend Acceptance.

===== FINAL UPDATE FROM DB Track PCs ====

The final decision for this paper has been taken by the program chairs after consultation with the SACs. All Senior Area Chairs have ranked papers according to the feedback from the AC during the review process. We decided to leave the original meta-review to reflect the opinion of the AC in light of the initial discussions with reviewers and SAC.